# Age-related and disease locus-specific mechanisms contribute to early remodelling of chromatin structure in Huntington's disease mice

Rafael Alcalá-Vida [1,2], Jonathan Seguin[1,2], Caroline Lotz[1,2], Anne M. Molitor[3,4,5,6], Ibai Irastorza-Azcarate[7], Ali Awada[1,2], Nezih Karasu [3,4,5,6], Aurélie Bombardier[1,2], Brigitte Cosquer[1,2], Jose Luis Gomez Skarmeta[8], Jean-Christophe Cassel[1,2], Anne-Laurence Boutillier[1,2], Thomas Sexton [3,4,5,6] & Karine Merienne [1,2]✉

Temporal dynamics and mechanisms underlying epigenetic changes in Huntington's disease (HD), a neurodegenerative disease primarily affecting the striatum, remain unclear. Using a slowly progressing knockin mouse model, we profile the HD striatal chromatin landscape at two early disease stages. Data integration with cell type-specific striatal enhancer and transcriptomic databases demonstrates acceleration of age-related epigenetic remodelling and transcriptional changes at neuronal- and glial-specific genes from prodromal stage, before the onset of motor deficits. We also find that 3D chromatin architecture, while generally preserved at neuronal enhancers, is altered at the disease locus. Specifically, we find that the HD mutation, a CAG expansion in the *Htt* gene, locally impairs the spatial chromatin organization and proximal gene regulation. Thus, our data provide evidence for two early and distinct mechanisms underlying chromatin structure changes in the HD striatum, correlating with transcriptional changes: the HD mutation globally accelerates age-dependent epigenetic and transcriptional reprogramming of brain cell identities, and locally affects 3D chromatin organization.

[1] Laboratoire de Neurosciences Cognitives et Adaptatives (LNCA), University of Strasbourg, 67000 Strasbourg, France. [2] CNRS UMR 7364, 67000 Strasbourg, France. [3] Institut de Genetique et de Biologie Moleculaire et Cellulaire (IGBMC), 67404 Illkirch, France. [4] CNRS UMR7104, 67404 Illkirch, France. [5] INSERM U1258, 67404 Illkirch, France. [6] University of Strasbourg, 67000 Strasbourg, France. [7] Berlin Institute of Medical Systems Biology (BIMSB), Max Delbrück Center for Molecular Medicine, Berlin, Germany. [8] Centro Andaluz de Biología del Desarrollo (CABD), CSIC-Universidad Pablo de Olavide-Junta de Andalucía, Seville, Spain. ✉email: karine.merienne@unistra.fr

Huntington's disease (HD) is a progressive inherited neurodegenerative disease caused by abnormal CAG repeat expansion in *HTT* coding region. While HD patients present average onset of motor symptoms at 35 years, subtle changes in behaviour and brain circuitry are observed at prodromal stage. Accordingly, the striatum, which is primarily affected in HD, undergoes early changes, altering brain connectivity[1,2]. As a result, it is believed that HD pathogenesis starts earlier than anticipated, which might have major therapeutic implication. However, we still lack early brain molecular correlates that would specify temporal dynamics of disease progression as well as provide insights into the mechanisms driving pathogenesis.

Epigenetic and transcriptional regulation are altered in HD brain tissues[3–11]. Particularly, in the striatum of HD patients and mice, neuronal identity genes are downregulated and depleted in H3K27 acetylation (H3K27ac), whereas glial-specific genes show an opposite trend[3,10]. However, it is unknown whether HD striatal epigenetic signatures progressively develop from early disease stage. Recent epigenomic and transcriptomic analyses using mouse tissues, including neural tissue, showed that variation in H3K27ac, a tissue-specific mark, is a key predictor of dynamic age-related transcriptional changes[12], which might suggest that H3K27ac changes at striatal identity genes in HD interfere with age-dependent mechanisms.

HD belongs to the family of short tandem repeat-associated diseases[13]. These unstable mutations are frequently located at boundaries of topological associated domains (TADs)[14]. It has been suggested that such chromatin architectural features might be hotspots for epigenetic misregulation, but it is yet unclear whether CAG expansion in the context of HD contributes to local remodelling of chromatin architecture.

Here, we defined temporal dynamics of epigenetic changes induced by the HD mutation, profiling the striatal epigenome of slowly progressing HD knockin (KI) mouse model at two early stages of pathology. Data integration with cell type-specific striatal enhancer and transcriptomic databases demonstrated acceleration of age-related epigenetic remodelling and transcriptional changes at neuronal- and glial-specific genes from the earliest stage, i.e. before the onset of motor deficits. Also, 3D chromatin architecture was generally preserved at neuronal-specific genes, though disrupted at selective loci. Specifically, CAG expansion impaired spatial organization of the chromatin in the region encompassing *Htt*, thereby affecting locally gene regulation. Collectively, we uncovered two early and distinct mechanisms underlying chromatin structure changes in HD striatum, and correlating with transcriptional changes: the HD mutation (1) globally accelerates age-dependent epigenetic and transcriptional reprogramming of brain cell identities, and (2) locally affects spatial organization of TADs adjacent to *Htt*.

## Results

### Early remodelling of epigenetic landscape in striatal neuron and glial cells of HD Q140 mice.
To define temporal dynamics of striatal epigenetic changes caused by the HD mutation, we used the Q140 line. HD Q140 heterozygous (het) mice have normal lifespan, display mild HD-like phenotypes and show limited neuronal death, even at late disease stage[15]. Motor function was preserved up to 6 months in Q140 het mice, though subtle behavioural changes reflecting prodromal stage were observed before 6 months (Supplementary Fig. 1). We used whole striatum of HD Q140 het and WT mice of 2 and 6 months to profile HD striatal epigenome at early disease stage, generating H3K27ac, H3K27me3 and RNA polymerase II (RNAPII) ChIP-seq data (Supplementary Fig. 2). HD is generally described as a sex-

independent disease affecting similarly males and females, though recent studies indicate sex-dependent effects influencing disease progression[16,17]. To avoid sex-dependent bias in our analyses, we used striatal tissue from both male and female mice. ChIPseq experiments using Q140 and WT samples of specific age and sex were performed simultaneously. For practical reasons, experiments performed on different sexes and at different ages were conducted at different times (see "Methods"). The data were of high quality, as shown by peak enrichment signal to noise rates, correlation analyses and additional quality analyses (Fig. 1a and Supplementary Figs. 2, 3). Moreover, principal component analysis (PCA) showed that sample variability was essentially explained by age (H3K27ac, RNAPII, H3K27me3), batch/sex (H3K27ac, RNAPII, H3K27me3) and genotype (H3K27ac) (Supplementary Fig. 3). Since sex and batch effects could not be distinguished, we focused our analyses on genotype- and age-dependent changes, analysing together male and female samples to assess epigenetic changes common to both sexes.

Differential enrichment analysis of ChIPseq data between Q140 and WT samples showed changes in H3K27ac from 2 months of age (Supplementary Fig. 4a). Regions depleted in H3K27ac in Q140 vs WT striata were enriched in gene ontology (GO) terms related to neuronal function, while regions showing increased H3K27ac in Q140 vs WT striata were enriched in terms implicated in glial function (Fig. 1b), suggesting distinct epigenetic signatures in neurons and glial cells establish from early disease stage. Regions differentially enriched in RNAPII between Q140 and WT samples displayed similar functional signatures, whereas H3K27me3 ChIPseq showed little changes between Q140 and WT striata (Fig. 1b and Supplementary Fig. 4a). Also, the amplitude of H3K27ac and RNAPII changes between Q140 and WT samples increased over time, demonstrating progressive nature of the mechanism (Supplementary Fig. 4a–c). Independent analysis of male and female ChIPseq samples supported the results of combined analysis of male and female samples (Supplementary Figs. 5, 6). Notably, H3K27ac was early depleted and enriched at neuronal and glial genes, respectively (Supplementary Fig. 5).

To specify temporal dynamics of epigenetic changes in HD neurons and glial cells, we generated H3K27ac and H3K27me3 ChIP-seq data in NeuN+ and NeuN− WT mouse striatal nuclei using the fluorescence activated nuclear sorting (FANS) approach[18] (Fig. 1c and Supplementary Figs. 2, 7a), which allowed the identification of neuronal and non-neuronal (essentially glial) specific enhancers (Supplementary Data 1). Integrated analysis of this cell type-specific striatal enhancer database with ChIPseq data generated on bulk striatum of Q140 and WT mice indicated that neuronal-specific enhancers were depleted in H3K27ac in Q140 vs WT samples, both at 2 and 6 months, whereas regions showing increased H3K27ac in Q140 vs WT samples predominantly originated from glial-specific enhancers (Fig. 1d, e). Remarkably, the effect was more specific at 2 vs 6 months (Fig. 1d, e and Supplementary Fig. 7b) and did not result from neuronal loss and/or astrogliosis, since the relative abundance of neuronal vs non-neuronal populations (including astrocytes) were comparable between the striatum of Q140 and WT mice (Supplementary Fig. 8). Thus, opposite remodelling of epigenetic landscape at neuronal- and glial-specific enhancers in HD mouse striatum is an early mechanism establishing at prodromal stage.

### Concomitant epigenetic and transcriptional reprogramming of striatal neuron and glial cell identities in HD Q140 mice.
We then investigated whether epigenetic and transcriptional changes were correlated in the striatum of HD Q140 mice, taking

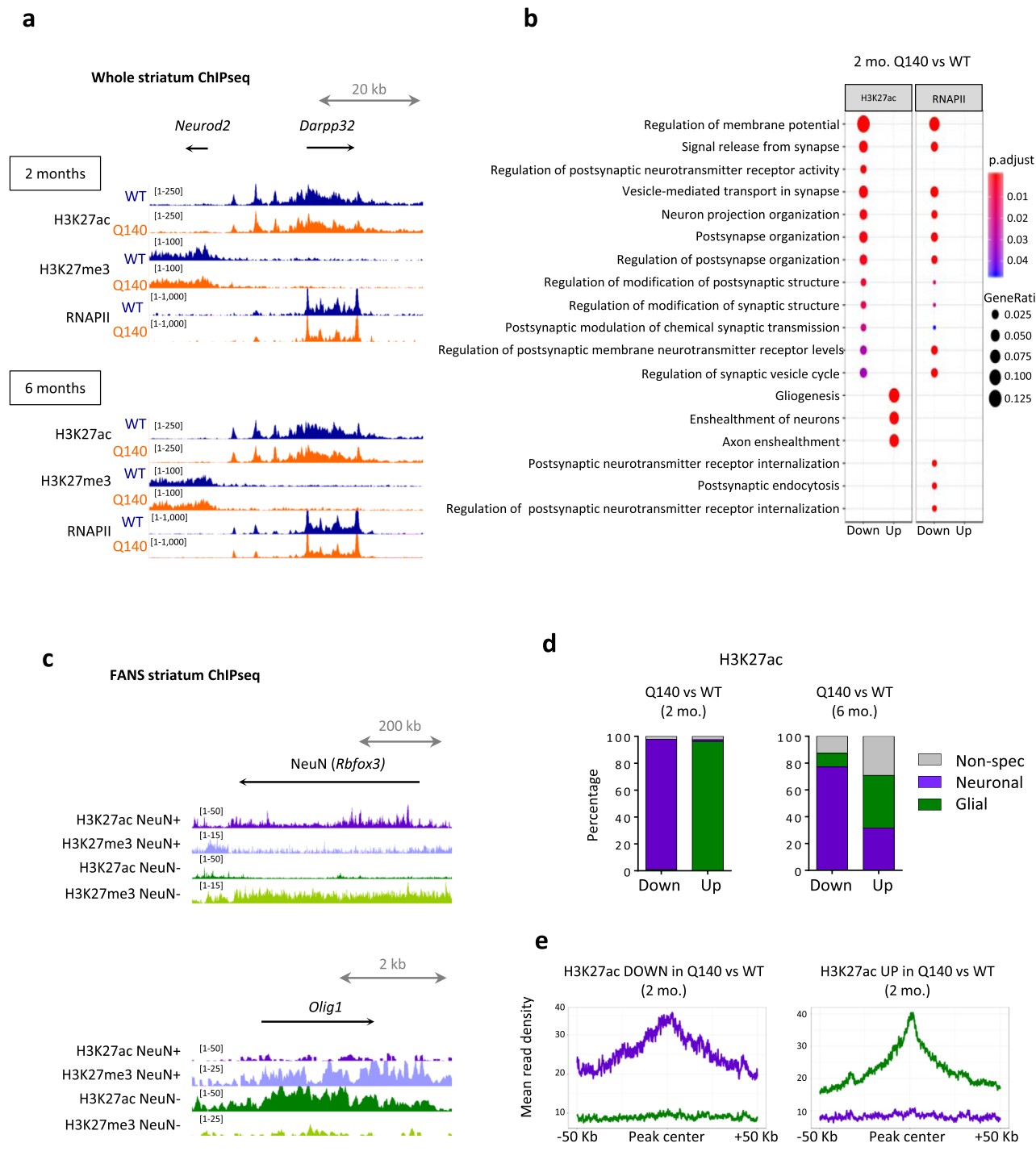

**Fig. 1 Striatal epigenetic alterations induced by the HD mutation establish early and in cell-type-dependent manner in HD Q140 mice. a** UCSC genome browser capture showing representative H3K27ac, H3K27me3 and RNAPII signals in the striatum of WT and Q140 mouse striatum at 2 and 6 months at selected locus, including active (*Darpp32* (*Ppp1r1b*)) and repressed (*Neurod2*) genes in the adult striatum. 2 mo., 2 months; 6 mo., 6 months **b** Gene Ontology analysis of regions differentially enriched in H3K27ac and RNAPII between Q140 and WT mouse striatal samples at 2 months (FDR < 0.05). Significant biological processes are shown using dot size proportional to gene ratio and heatmap reflecting adjusted *P* value. **c** UCSC genome browser capture showing representative H3K27ac and H3K27me3 signals in striatal NeuN+ and NeuN− populations in WT mice at 6 months at selected neuronal gene (*NeuN* (*Rbfox3*)) and glial gene (*Olig2*). **d** Bargraphs showing cell-type distribution of regions differentially enriched in H3K27ac in Q140 vs WT mouse striatum at 2 and 6 months of age. **e** Metaprofiles showing H3K27ac signal in NeuN+ and NeuN− sorted nuclei, considering differentially enriched peaks in Q140 vs WT striata at 2 months. Source data are provided as a Source Data file.

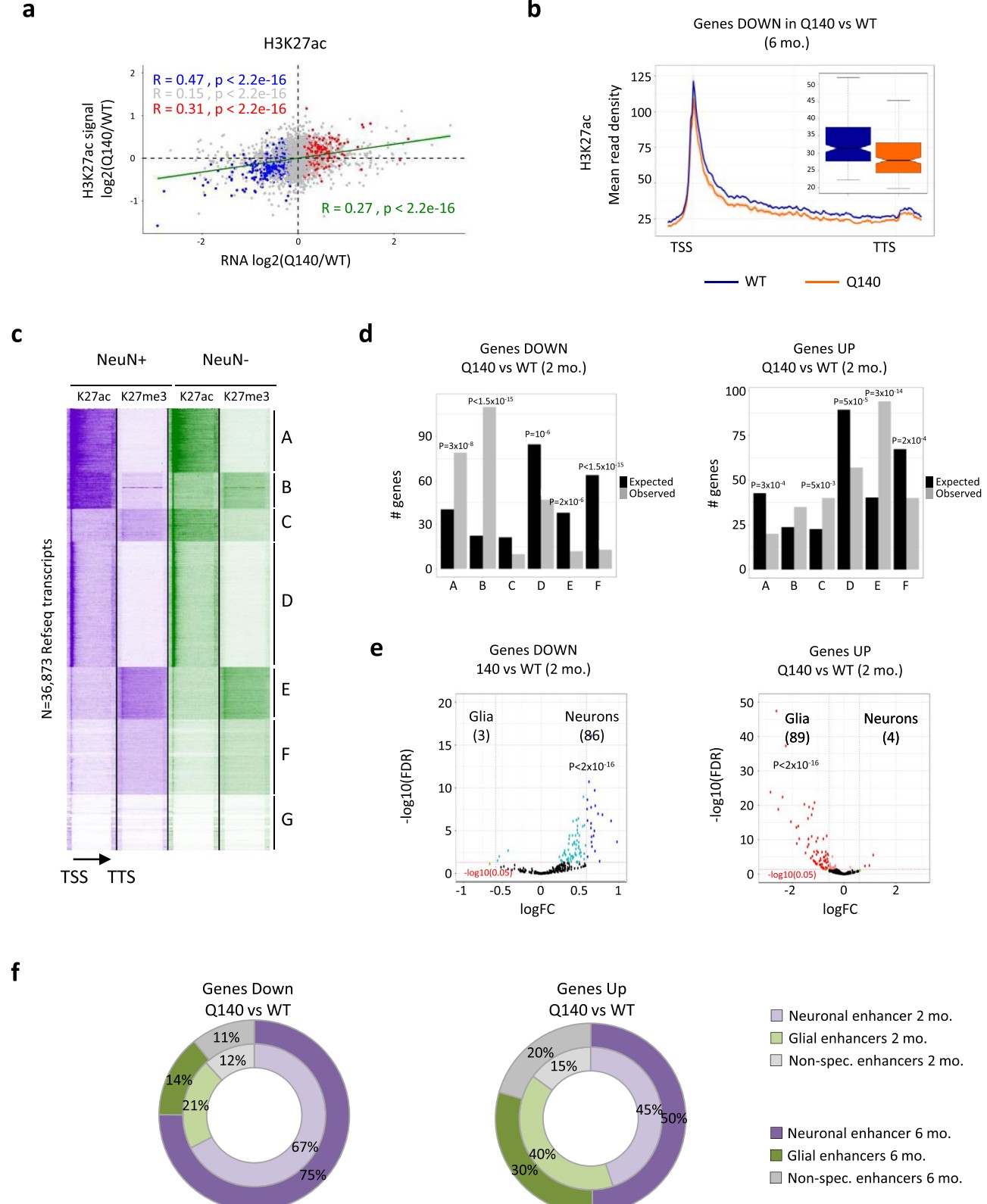

advantage of RNAseq data on HD KI mice[5] (Supplementary Fig. 9a). Integrated analysis was performed between H3K27ac ChIPseq data generated on 2- and 6-month-old striatal tissue of Q140 and WT mice and RNAseq data generated on same mouse line, tissue and ages. Linear regression analysis showed that H3K27ac and mRNA changes between Q140 and WT samples were significantly and positively correlated, with a correlation

stronger for down- than up-regulated genes (Fig. 2a). Gene metaprofile analysis supported the conclusion, showing that downregulated genes in Q140 vs WT striatum were specifically depleted in H3K27ac (Fig. 2b and Supplementary Fig. 9b). Similar trends were observed between RNAPII and mRNA changes, but not between H3K27me3 and mRNA changes (Supplementary Fig. 9c, d).

**Fig. 2 Concomitant epigenetic and transcriptional reprogramming of neuronal- and glial-specific genes in HD Q140 striatum. a** Linear regression analysis between transcriptional and H3K27ac changes in the striatum of Q140 vs WT mice of 6 months. The correlation is shown for all genes (green), genes significantly downregulated (Fold change (FC) <1 and adj. $P$ value <0.05; blue), genes significantly upregulated (FC >1 and adj. $P$ value <0.05; red) and non-significantly altered genes (grey). Pearson's correlation index and $P$ value for fitted linear model are shown. **b** Gene body metaprofiles representing H3K27ac read count distribution for top 300 downregulated genes, ranked according to adj. $P$ value, in Q140 mouse striatum at 6 months. TSS transcription start site; TTS transcription termination site. Data from male and female samples were used to generate average profile. Boxplots represent the distribution of mean read density along the profiles and show median, first quartile (Q1), third quartile (Q3) and range (min, $Q1-1.5\times(Q3-Q1)$; max, $Q3+1.5\times(Q3-Q1)$). **c** Heatmap of the 36,873 annotated mm10 RefSeq gene transcripts, integrating H3K27ac and H3K27me3 gene profiles from NeuN+ and NeuN− sorted nuclei and showing seven distinct epigenetic profiles generated by $k$-means clustering (clusters A-G). The arrow indicates the orientation of genes; TSS transcription start site; TTS transcription termination site. **d** Histograms showing cluster distribution of genes down- (upper panel) and upregulated (lower panel) in Q140 vs WT striatum at 2 months of age. Three-hundred top dysregulated genes were analysed from RNAseq data[5] and ranked according to $P$ value. Observed numbers were compared with expected numbers and a binomial test (two-sided) was used to assess significant differences, with multiple testing correction using the Bonferroni method. **e** Volcano plot representation of differential expression values between glial cells (astrocytes and microglia) and neurons (medium spiny neurons, MSNs, including D1 and D2 MSNs) using top-ranked 300 genes down (top) and 300 genes up (bottom) in Q140 vs WT striatum at 2 months. Genes down in Q140 vs WT striatum and significantly changed in neurons vs glial cells (FC >1 and adj. $P$ value <0.05) are shown in blue; genes up in Q140 vs WT striatum and significantly changed in neurons vs glial cells (FC <1 and adj. $P$ value <0.05) shown in red. A binomial test (two-sided) was performed to assessed enrichment in neuronal- or glial-specific genes. Adjustment for multiple comparisons was not performed. **f** Pie chart showing the distribution of neuronal-, glial- and non-specific enriched H3K27ac regions (as defined in Supplementary Dataset 1) associated with top 300 genes down (left) and top 300 genes up (right) in Q140 vs WT striatum at 2 and 6 months. Source data are provided as a Source Data file.

Neuronal identity genes, regulated by super enhancers consisting in broad enhancers highly enriched in H3K27ac throughout their target genes[19], are downregulated and depleted in H3K27ac in HD striatum, whereas glial identity genes display an opposite trend[3,10]. To investigate early epigenetic signatures of dysregulated genes in HD mouse striatum at cell type-specific levels, we delineated neuronal and non-neuronal (glial) super enhancer-regulated genes using H3K27ac and H3K27me3 ChIPseq data generated on WT NeuN+ and NeuN− striatal nuclei (Fig. 1c, Supplementary Fig. 2 and Supplementary Data 1). Using $k$-means clustering, we retrieved seven distinct clusters, including clusters enriched in neuronal and non-neuronal super enhancer-regulated genes, respectively (clusters B and C) (Fig. 2c). Specifically, cluster B, which contained neuronal super enhancer-regulated genes, was enriched in H3K27ac and depleted in H3K27me3 in NeuN+ nuclei, but depleted in H3K27ac and enriched in H3K27me3 in NeuN− nuclei (Fig. 2c). Cluster C, comprising non-neuronal super enhancer-regulated genes, showed opposite features (Fig. 2c). Additionally, we identified a cluster enriched in H3K27ac and depleted in H3K27me3 in both NeuN+ and NeuN− nuclei, likely containing super enhancer-regulated genes common to neurons and non-neuronal cells (cluster A, Fig. 2c). Consistently, clusters B and C were enriched in neuronal- and glial-specific genes, respectively, and cluster A contained similar proportion of neuronal- and glial-specific genes (Supplementary Fig. 10a). As expected, genes in clusters B and C were enriched in GO terms reflecting neuronal and glial identities, respectively, while terms associated with cluster A were less homogenous and linked to nucleosome assembly, cell adhesion, and energy metabolism (Supplementary Fig. 10b). Importantly, downregulated genes in the striatum of Q140 vs WT mice at 2 and 6 months were most significantly enriched in the neuronal super enhancer cluster (cluster B, Fig. 2d and Supplementary Fig. 10c), demonstrating that neuronal identity genes are early prone to downregulation in HD mouse striatum. In contrast, upregulated genes in Q140 striatum at 2 months of age were significantly enriched in the glial identity gene cluster (cluster C), as well as in cluster E, containing developmental genes enriched in glial-specific genes (Fig. 2d and Supplementary Fig. 10a, b). Together, these results suggest that maintenance of neuronal and glial identities in the striatum is early challenged by the HD mutation.

Supporting this view, integration of Q140 RNAseq data[5] with cell type-specific striatal transcriptomic database[10] showed that GABAergic medium spiny neurons (MSNs), predominant in the striatum and primarily affected in HD, were highly enriched in downregulated genes in Q140 vs WT striatum, particularly at 2 months, while glial cells were enriched in upregulated genes (Fig. 2e and Supplementary Fig. 10d, e). Finally, integration of Q140 RNAseq data[5] with cell type-specific enhancer database (Supplementary Fig. 2, Supplementary Data 1) showed that downregulated genes in Q140 vs WT striatum were predominantly associated with neuronal-specific enhancers, while substantial glial-specific enhancers were more associated with upregulated genes (Fig. 2f). Together, these results indicate that early remodelling of neuronal- and glial-specific enhancers in HD striatum correlates with transcriptional changes at neuronal and glial identity genes.

**Age-related transcriptional changes at striatal identity genes are accelerated in HD Q140 mice.** Variation in the amount of H3K27ac at enhancers is a key predictor of age-related transcriptional changes[12], suggesting that age might interact with the HD mutation during epigenetic and transcriptional reprogramming of striatal cell identities. To explore this hypothesis, we first assessed age-dependent transcriptional changes of neuronal- and glial-specific genes in Q140 and WT striatum. Remarkably, neuronal-specific genes were enriched in genes whose expression decreased with age in WT striatum, whereas glial-specific genes showed an opposite trend (Fig. 3a). Importantly, physiological age-dependent transcriptional changes of neuronal- and glial-specific genes were accelerated by the HD mutation: neuronal-specific genes, most particularly genes specific to MSN expressing D1 dopamine receptor (D1 MSN), including *Drd1*, were lower in Q140 than in WT striata at both ages, whereas glial-specific genes (e.g. *Tmem151b*[20]), were increased in Q140 vs WT striatum (Fig. 3b and Supplementary Fig. 11a, b). Integration of transcriptomic databases[5,10] with epigenetic clustering generated in Fig. 2c further supported the conclusions, showing that down-regulation of neuronal identity genes (genes in cluster B) and upregulation of glial identity genes (genes in cluster C) were accelerated in Q140 striatum (Supplementary Fig. 12). Thus, neuronal and glial identity genes are regulated in opposite directions with age in mouse, and the HD mutation exacerbates those age-related transcriptional changes.

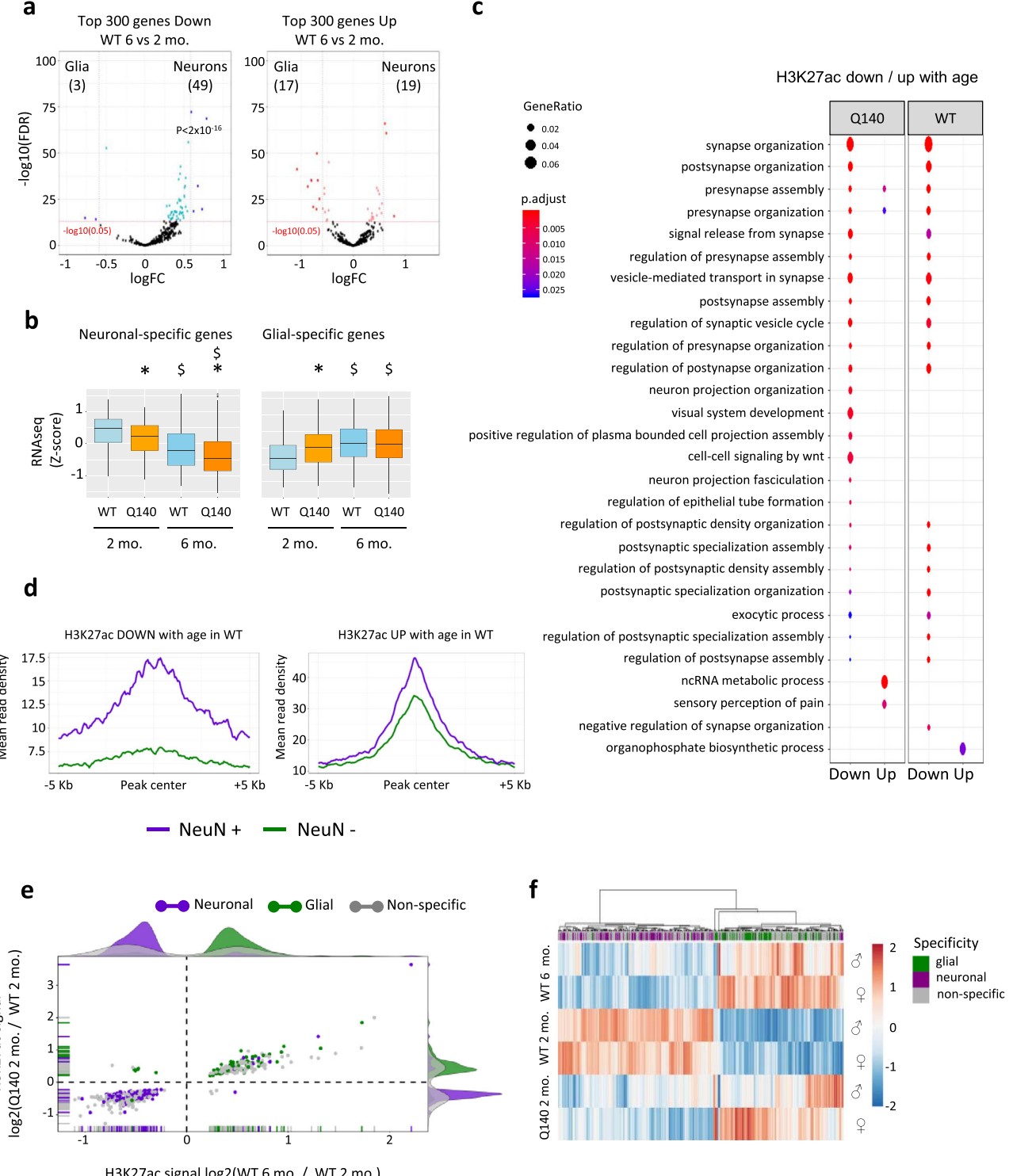

**Age-related epigenetic remodelling of neuronal- and glial-specific enhancers is accelerated in HD Q140 mice**. We then determined whether acceleration of age-related transcriptional changes in HD mouse striatum would associate with age-related epigenetic mechanisms. Age was the major component of variability of H3K27ac, RNAPII and H3K27me3 ChIPseq data generated on Q140 and WT striata at 2 and 6 months of age (Supplementary Fig. 13a). Moreover, GO analysis indicated that regions depleted in H3K27ac with age associated with neuronal functions in both WT and Q140 samples, whereas functional signatures of regions increased in H3K27ac with age were less

consistent (Fig. 3c). Analysis of age-dependent RNAPII changes led to similar signatures, while few GO terms reflecting H3K27me3 changes with age were significantly enriched, whether in WT or Q140 samples (Supplementary Fig. 13b, c). These results suggest that neuronal-specific genes are particularly prone to reduced H3K27ac occupancy and RNAPII recruitment with age. To further explore this possibility, H3K27ac ChIPseq data generated at 2 and 6 months were integrated with cell type-specific enhancer database (Supplementary Fig. 2, Supplementary Data 1). Metaprofile analysis showed alteration of neuronal-specific enhancers over time (Fig. 3d). Specifically, regions

**Fig. 3 Age-related epigenetic and transcriptional reprogramming of neuronal and glial identities are accelerated in the striatum of HD Q140 mice.**
**a** Volcano plot representation of differential expression values between glial cells (astrocytes and microglia) and neurons (medium spiny neurons, MSNs, including D1 and D2 MSNs) using top-ranked (according to adj. $P$ val) 300 genes down (left) and top-ranked 300 genes up (right) in WT striatum at 6 months vs 2 months. Genes down at 6 vs 2 months in WT striatum and significantly changed in neurons vs glial cells (FC >1 and adj. $P$ value <0.05) are shown in blue; genes up at 6 vs 2 months in WT striatum and significantly changed in neurons vs glial cells (FC <1 and adj. $P$ value) are shown in red. A binomial test (two-sided) was performed to assess enrichment in neuronal- or glial-specific genes. Adjustment for multiple comparisons was not performed. **b** Boxplots representing $z$-score values computed from RNAseq data generated in Q140 and WT striatum at 2 months and 6 months, considering genes increased in neurons vs glial cells (neuronal-specific genes, left) and genes increased in glial cells vs neurons (glial-specific genes, right). Boxplots show median, first quartile (Q1), third quartile (Q3) and range (min, Q1−1.5×(Q3−Q1); max, Q3+1.5×(Q3−Q1). Statistical analysis was performed using Kruskal–Wallis test (one-sided), with multiple testing correction using the Benjamini-Hochberg method. Neuronal-specific genes: *, $P < 2 \times 10^{-16}$, Q140 vs WT comparison at 2 months; *, $P < 2 \times 10^{-16}$, Q140 vs WT comparison at 6 months; $, $P < 2 \times 10^{-16}$, 6- vs 2-month comparison in WT; $, $P = 2 \times 10^{-13}$, 6- vs 2-month comparison in R6/1. Glial-specific genes: *, $P < 2 \times 10^{-16}$, Q140 vs WT comparison at 2 months; *; $, $P < 2 \times 10^{-16}$, 6- vs 2-month comparison in WT; $, $P = 9 \times 10^{-10}$, 6- vs 2-month comparison in R6/1. RNAseq data from transcriptomic databases[5,10] were used for these analyses. **c** Gene Ontology analysis of regions differentially enriched in H3K27ac in 6- vs 2-month striatal samples, in Q140 and WT contexts (FDR < 0.05). Significant biological processes are shown using dot size proportional to gene ratio and heatmap reflecting adj. $P$ value. **d** Metaprofiles showing H3K27ac signal in NeuN+ and NeuN− sorted nuclei, considering differentially enriched peaks in WT striatal samples of 6 vs 2 months. FC <1 and adj. $P$ value <0.05, down, left; FC >1 and adj. $P$ value <0.05, up, right. **e** Scatter plot and density population graphs representing log2 of fold-change in H3K27ac at regions significantly changed ($P < 0.05$) both in Q140 vs WT samples at 2 months and in WT samples at 6 vs 2 months. Differentially H3K27ac-enriched regions distribute in three categories: non-specific (Non-specific, grey), neuronal-specific (Neuronal, purple) and glial-specific (Glial, green). **f** Heatmap representing $z$-score values of H3K27ac signal at regions differentially enriched in H3K27ac ($P < 0.05$) both in Q140 vs WT samples at 2 months and in WT samples at 6 vs 2 months. Differentially H3K27ac-enriched regions distribute in three categories: non-specific (Non-specific, grey), neuronal-specific (Neuronal, purple) and glial-specific (Glial, green); hierarchical clustering was performed according to H3K27ac signal. Source data are provided as a Source Data file.

depleted in H3K27ac with age in normal striatum were enriched in neuronal-specific enhancers. In contrast, regions increased in H3K27ac with age originated from both glial- and neuronal-specific enhancers (Fig. 3d).

To investigate possible effect of the HD mutation on age-dependent regulation of neuronal- and glial-specific enhancers, we performed linear regression analysis, comparing the effects of genotype and age on H3K27ac levels: the effects of genotype and age on H3K27ac were significantly and positively correlated, at both neuronal- and glial-specific enhancers (Supplementary Fig. 14a). More specific analysis revealed that regions significantly depleted in H3K27ac with age were also significantly depleted in H3K27ac by the HD mutation, and predominantly originated from neuronal-specific enhancers, whereas glial-specific enhancers contributed to regions showing significant increased H3K27ac with age and in response to the HD mutation (Fig. 3e). Furthermore, hierarchical heatmap representation of regions showing significant H3K27ac variations in both age- and genotype-dependent manners supported this conclusion. Neuronal- and glial-specific enhancers essentially distributed in two distinct clusters, with the cluster of neuronal-specific enhancers containing regions depleted in H3K27ac with age and by the HD mutation, and the cluster of glial-specific enhancers comprising regions with opposite age- and genotype-dependent H3K27ac variations (Fig. 3f).

Moreover, predominant neuronal origin of striatal regions depleted in H3K27ac upon age and in response to the HD mutation was supported by DNA motif analysis, showing enrichment in motifs recognized by DLX/GATA and GCM1/2 (Supplementary Fig. 14b), which are transcription factors essential to the establishment of neuronal fate and striatal identity[21–25]. In contrast, glial origin of striatal regions showing increased H3K27ac due to age and the HD mutation was consistent with enrichment in DNA motif binding THAP12 (Supplementary Fig. 14b), implicated in inflammation and stress response[26]. Finally, neuronal-specific enhancers depleted in H3K27ac in age- and genotype-dependent manners associated with complex network of co-regulated genes, which was significantly enriched in GO terms related to neuronal functions (Supplementary Fig. 14c, d). In contrast, poor complexity

characterized the network of co-regulated genes associated with glial-specific enhancers showing age- and HD mutation-dependent increase in H3K27ac (Supplementary Fig. 14c), suggesting a reduced impact of the HD mutation on H3K27ac regulations in glial- than neuronal-specific genes. Together, these results indicate that the HD mutation early accelerates age-dependent remodelling of chromatin landscape at striatal enhancers, precipitating depletion in H3K27ac at neuronal-specific enhancers and, to lesser extent, exacerbating H3K27ac enrichment at glial-specific enhancers, thereby resulting in acceleration of age-related regulation of neuronal and glial identity genes.

**3D chromatin architecture is selectively impaired at striatal identity genes in HD mice.** To further define the relationship between epigenetic and transcriptional changes in Q140 striatum, we performed 3D chromatin architecture analyses. High levels of H3K27ac at super-enhancers correlate with extensive chromatin looping between promoters and enhancers, which facilitates rapid transcription dynamics[27–29]. We hypothesized that dysregulation of super enhancer-regulated genes in HD striatum may associate with changes in spatial organization of the chromatin, and generated 4C-seq data to explore this possibility. Experiments were performed using the striatum of male and female Q140 and WT mice at 6 months of age, and targeting super enhancer-regulated genes downregulated in HD striatum, including *Pde10a*, *Gpr6* and *Ptpn5*, and non-super enhancer-regulated genes such as *Msh2*, as a control (Fig. 4a, b, Supplementary Figs. 15, 16a–c and Supplementary Data 2). Chromatin looping at *Pde10a* was impaired in male and female Q140 samples (Fig. 4b and Supplementary Fig. 15). Specifically, we identified striatal interactions of the *Pde10a* promoter with four upstream H3K27ac-enriched regions, and three intronic regions of the gene. The upstream interactions were consistently reduced, and intronic interactions were increased, in Q140 vs WT mice. Similar results were obtained from the analysis of 4C-seq data generated on the striatum of another HD mouse, the HD R6/1 transgenic model, over-expressing *HTT* exon-1 with CAG expansion[30] (Fig. 4c). Furthermore, upstream interacting regions were not observed in embryonic stem cells (mESC)[31], suggesting a role in cell type-specific expression of *Pde10a* (Fig. 4c). Finally, H3K9me3

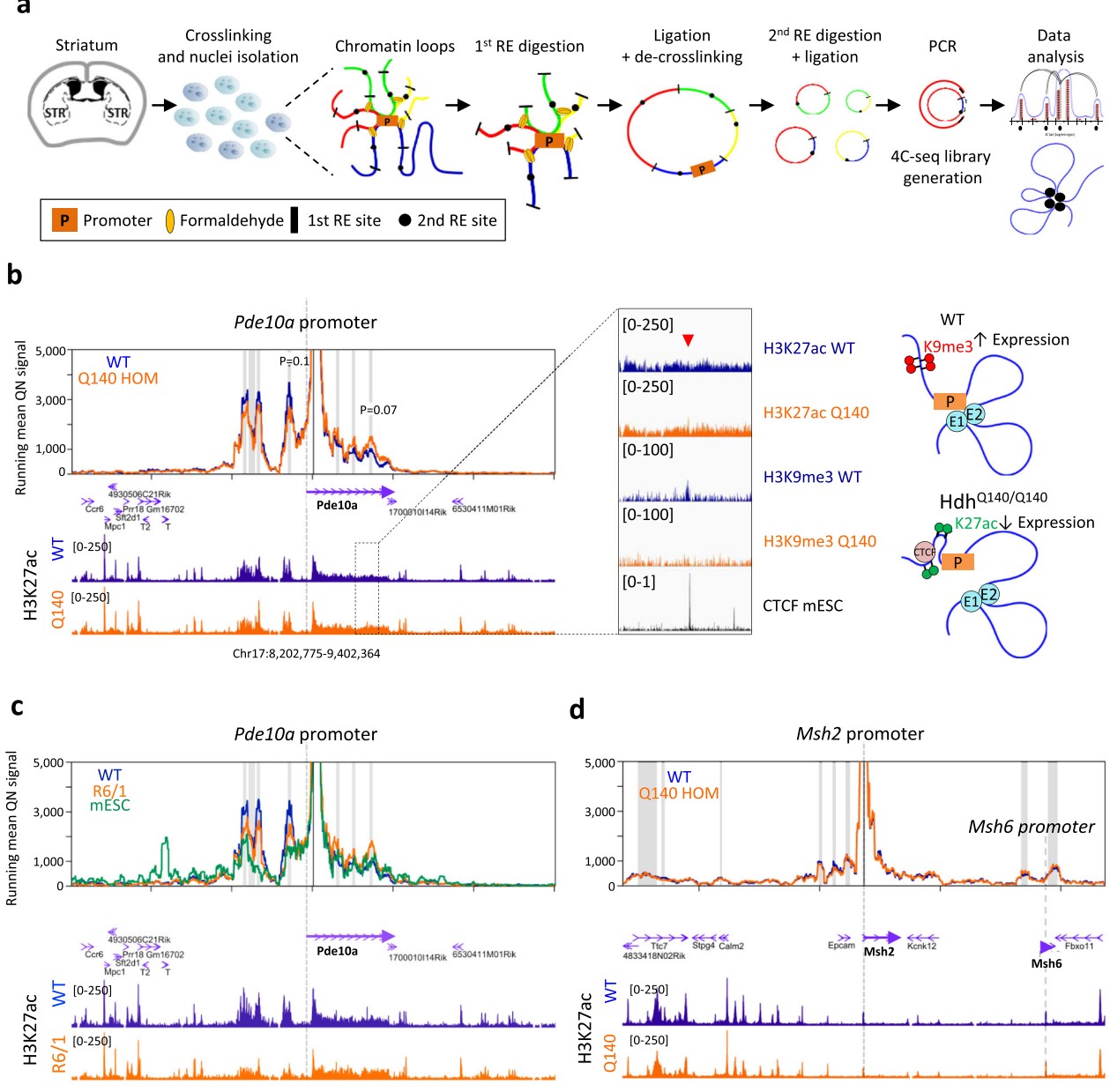

**Fig. 4 Chromatin architecture at _Pde10a_ is impaired in the striatum of HD Q140 mice. a** Scheme showing 4C-seq technique major steps using mouse striatum. PCR primers specific to each bait can be found in the "Methods" section. RE restriction enzyme. **b** On the left, 4C-seq profiles at _Pde10a_ locus using HD Q140 (orange) and WT (blue) mouse striatum at 6 months. The mean of male and female 4C-seq quartile normalized read counts is plotted as the main lane for each condition. Statistical analysis of differential interacting peaks in Q140 vs WT was performed using two-paired _t_-test, with multiple testing correction using the Benjamini-Hochberg method. _P_ = 0.1 (upstream of _Pde10a_ promoter), _P_ = 0.07 (downstream of _Pde10a_ promoter). Gene annotations are included as well as H3K27ac ChIPseq signals (using ChIPseq data generated in this study on the striatum of Q140 and WT mice at same age). Grey shadows show specific interacting regions. On the middle, zoom into _Pde10a_ intronic region, showing H3K27ac and H3K9me3 levels in WT and Q140 mice striatum together with CTCF enrichment (from CTCF ChIP-seq data generated in mESC). On the right, model to explain chromatin conformational changes at _Pde10a_ locus in HD mouse striatum. **c** 4C-seq profiles at _Pde10a_ locus generated using HD R6/1 (orange) and WT (blue) mouse striatum at 14 weeks of age and using mESC (green). Gene annotations and H3K27ac ChIPseq signals[3] are included. Grey shadows show specific interacting regions. **d** 4C-seq profiles at _Msh2_ locus using Q140 (orange) and WT (blue) striatum at 6 months. The mean of male and female 4C-seq quartile normalized read counts is plotted as the main lane for each condition. Gene annotations and H3K27ac ChIPseq signals are included. Grey shadows show specific interacting regions. _Msh6_ promoter is annotated to highlight the distal chromatin loop formed with _Msh2_ promoter.

ChIPseq data generated on WT mouse striatum showed a specific H3K9me3 peak located at a _Pde10a_ intronic region, which interacts with the promoter, coinciding with a CTCF peak (Fig. 4b and Supplementary Fig. 16d, e). Convergent CTCF motifs are a hallmark of many chromatin loops[32], and the _Pde10a_ promoter and interacting intronic region comprise such a convergent CTCF pair. Remarkably, this H3K9me3 intronic peak was absent in Q140 striatum (Fig. 4b), suggesting that a repressive intragenic _Pde10a_ loop may be stabilized at the expense of upstream activating loops in HD striatum (Fig. 4b), a mechanism reminiscent to the regulation of chromatin architecture at _Grin2b_[33].

Analysis of additional loci indicated that the HD mutation did not substantially affect chromatin looping at the other tested super enhancer-regulated genes or at *Msh2* gene, though specific promoter–enhancer interactions were observed for these genes (Fig. 4d, Supplementary Fig. 16a, b and Supplementary Data 2). For example, we identified a chromatin loop between *Msh6* and *Msh2* promoters (Fig. 4d), which might explain the co-regulation of these two DNA repair proteins forming a heterodimer[34]. Altogether, these results suggest that chromatin topology is largely unchanged by H3K27ac depletion at neuronal super enhancers or transcriptional downregulation of their target genes during HD onset, although locus-specific architectural changes involving additional mechanisms may be observed at subsets of neuronal super enhancers.

**CAG expansion locally affects 3D chromatin architecture and transcription in the striatum of HD Q140 mice.** Disease-associated short tandem repeats, including CAG expansion in *HTT*, are often located to TAD boundaries[14]. In the case of Fragile X, CGG expansion at *FMR1* was shown to alter proximal chromatin topology. We hypothesized that CAG expansion in the context of HD might similarly impair chromatin architecture of TADs encompassing *HTT*. Consistent with highly conserved TAD organization between mice and humans[35], CAG expansion in human and murine *HTT* is located in the vicinity of a TAD border as depicted by human hippocampal[36] and mouse cortical neuronal[37] Hi-C data (Fig. 5a). We therefore addressed the hypothesis using the striatum of Q140 homozygous (hom) mice (and WT mice as controls), especially since Q140 hom mice, presenting both alleles with CAG expansion in its proper genomic context, embody an ideal experimental model. Striatal tissues of both male and female animals were used in the analysis. Specifically, using striatal tissue of Q140 hom and WT mice, we generated 4C-seq data targeting 5 promoter regions of genes encompassing a two-megabase window of the genome, comprising *Htt* (i.e. *Mxd4, Nop14, Htt, Lrpap1* and *Acox3*) (Fig. 5b). Then, by using the recently developed 4Cin bioinformatics tool[38], we modelled three-dimensional chromatin architecture of this two-megabase region in HD and WT genomic contexts (Fig. 5c and Supplementary Fig. 17a–c).

Overall, generated 3D models showed high levels of similarity across datasets, with correlation coefficients ranging from 0.8 to 0.87 (Supplementary Fig. 17a). Remarkably, superposition between those models and our H3K27ac, H3K27me3 and RNAPII ChIPseq data (Supplementary Fig. 17b and Supplementary Movies 1, 2) indicated that compacted regions of 3D models were enriched in H3K27me3 and depleted in H3K27ac and RNAPII, whereas, on the opposite, open regions were enriched in H3K27ac and RNAPII and depleted in H3K27me3. Moreover, transcriptional activity of genes located in the region was related to spatial proximity of H3K27ac, H3K27me3 and RNAPII ChIPseq peaks (Supplementary Fig. 17c). These analyses showing high degree of coherence across chromatin spatial organization predicted by 3D models, epigenetic features assessed by ChIPseq data analysis, and transcriptional rate measured from RNAseq data, provided strong support to 3D model validity.

To evaluate possible alteration in TAD organization in Q140 mice, we computed an insulation score, quantifying TADs and subTADs boundaries[39]. Regions at TADs/subTADs boundaries associate with low level of compaction and, in consequence, low insulation score. Consistently, *Htt* region in WT striatum was located at TAD boundary, since it associated with low insulation score (Fig. 5c, green arrows). Remarkably, however, corresponding insulation score in Q140 samples was increased, most prominently for female sample. Finally, re-organization of subTAD boundary

upstream to *Htt* was consistently observed in Q140 vs WT striata (Fig. 5c, black arrows). In Q140 samples, this boundary was displaced to a downstream region when compared to WT samples. More specifically, the boundary was located <1.5 Mb and >1.5 Mb with respect to distance scale in Q140 and WT samples, respectively (Fig. 6c, black arrows). Together, these results indicate that CAG expansion at *Htt* locally impairs TAD insulation.

Moreover, re-analysing RNAseq data generated in the striatum of Q140 and WT mice, we observed that differentially expressed genes (DEGs) were significantly enriched on chromosome 5, containing *Htt* (Fig. 5d). At 2 months, 16% of DEGs were located in chromosome 5 and they were predominantly in *Htt* close neighbourhood (Fig. 5d). *Htt* itself and juxtaposed gene *Grk4* were decreased and increased in the striatum of HD KI mice, respectively (Supplementary Fig. 17d). As expected, chromosome 5 was not enriched in DEG in the striatum of R6/1 transgenic mice overexpressing CAG-expanded *HTT* exon-1 vs WT mice[30,40], and *Htt* and *Grk4* were unchanged in the striatum of R6/1 mice (Supplementary Fig. 17d–f). Together, this indicates that CAG expansion at *Htt* in striatal tissue locally impairs spatial genome organization, which in turn likely affects transcriptional regulation of the region.

## Discussion

Here we have used slowly progressing KI mouse model of HD to investigate HD-associated epigenetic signatures at early disease stages. We have generated H3K27ac, RNAPII and H3K27me3 ChIPseq datasets using striatal tissues of HD mice and WT mice at 2 and 6 months, corresponding to prodromal stages. We have also generated H3K27ac and H3K27me3 ChIPseq datasets in neuronal and non-neuronal striatal cell populations, as well as striatal 4C-seq data at selected genes, including neuronal identity genes (regulated by super-enhancers) and 4C-seq data encompassing *Htt* locus. Those epigenomic datasets were integrated with transcriptomic databases, generated with same mouse model, and with cell-type specific transcriptomic databases[5,10]. We show that in HD mouse striatum, epigenetic and transcriptional reprogramming is an age-related mechanism establishing from prodromal stages: the HD mutation exacerbates age-dependent decreased and increased H3K27ac at neuronal- and glial-specific enhancers, respectively, with RNAPII and gene expression changes following similar dynamics. Thus, we provide evidence that the HD mutation leads to acceleration of age-related transcriptional and epigenetic regulation of neuronal and glial cell identities in the striatum. Moreover, our analyses focused on *Htt* locus reveal local rearrangements of 3D chromatin architecture encompassing CAG expansion, which likely contributes to local dysregulation of transcription. Collectively, these results indicate that epigenetic and transcriptional signatures in HD striatum comprise two distinct components, which are age-related and disease-locus specific.

Epigenetic aging in the brain is an emerging concept that is still poorly defined. The recent studies showing that few hundreds of DNA methylation sites can be used to assess the "epigenetic clock" of a given tissue represent few exceptions[41,42]. Analysing DNA methylation from post-mortem brain tissues of HD patients and control individuals, Horvath and collaborators observed a positive correlation between epigenetic age and HD status, which suggests that the HD mutation leads to acceleration of aging[41]. However, such a correlation was not measured when analysing the striatum, possibly due to major striatal neuronal loss in the samples analysed, collected at rather late disease stage. Our study showing that H3K27ac may be used as a biomarker to assess epigenetic aging extends these results and suggest that the HD

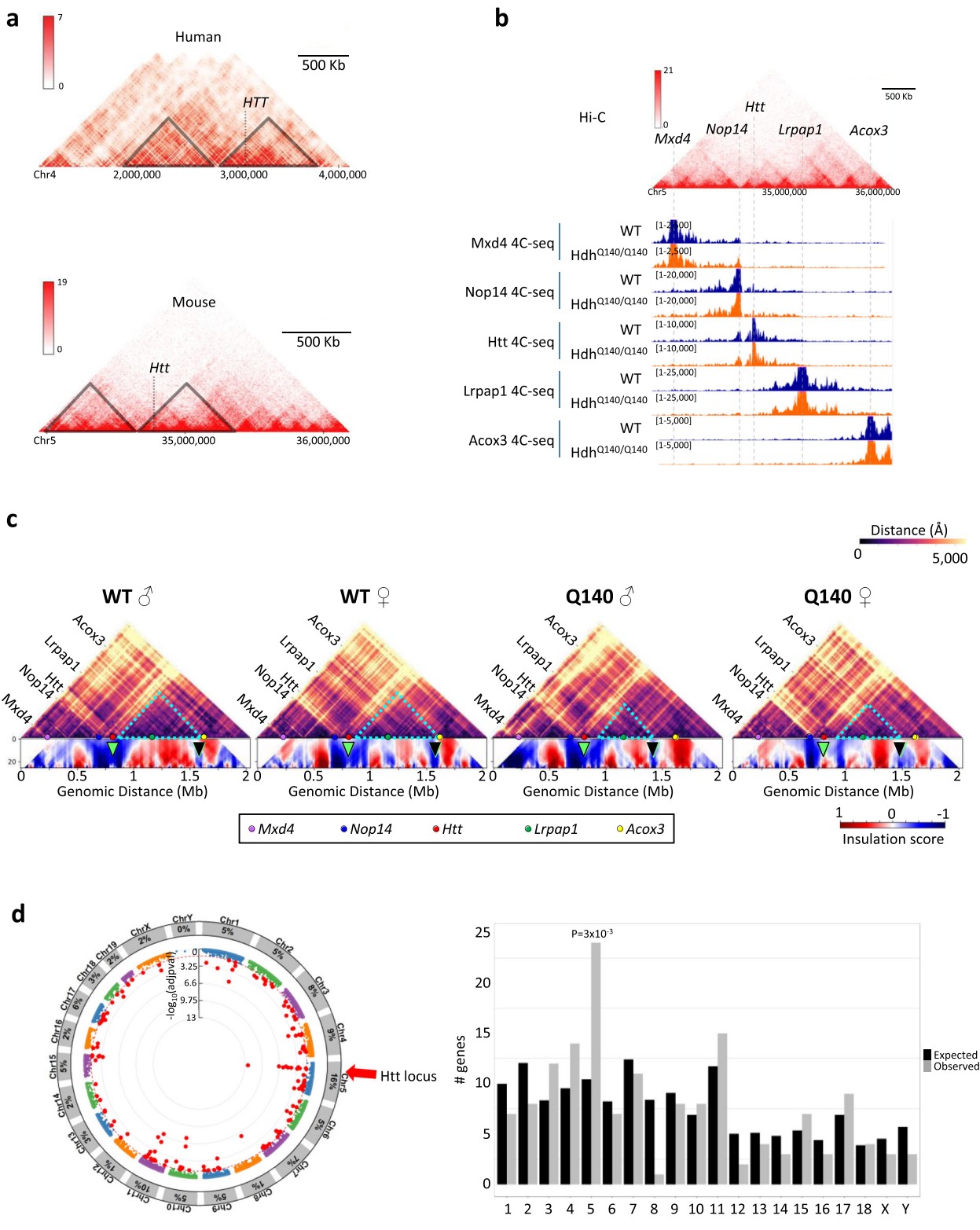

striatum, which is primarily affected in HD, undergoes accelerated epigenetic aging. When precisely acceleration of epigenetic aging may start in HD striatum will remain an open question, but the finding that it might be observed far before motor symptoms arise in HD KI mice indicates that it could be an early biomarker.

Investigating neuronal and non-neuronal (essentially glial) enhancers in mouse striatum using H3K27ac, a cell type-specific

histone modification, we provide evidence for age-dependent epigenetic reprogramming of brain cell identities, and show that the mechanism correlates with the transcriptional response. Remarkably, the direction of age-related epigenetic and transcriptional changes at neuronal and glial identity genes was opposite, and consistent with major features of brain aging, i.e. progressive alteration of neuronal activity and activation of glial

**Fig. 5 The HD mutation induces disease locus-specific alterations of chromatin architecture and transcription regulation in the striatum of Q140 mice.**
**a** Hi-C data capture showing 3 and 2.3 megabase genome region from human hippocampus and mouse cortical neurons, respectively. In both cases, *Htt* is in the vicinity of TAD borders (~250 Kb for human *HTT* and ~115 Kb for mouse *Htt*). **b** Genome browser representation of mouse cortical neuron Hi-C data, zooming on the region encompassing *Htt*, and aligned with 4C-seq data generated in this study using striatal tissue of WT (blue) and Q140 (orange) mice. The genomic locations of *Mxd4*, *Nop14*, *Htt*, *Lrpap1* and *Acox3* 4C-seq baits are indicated. **c** Virtual Hi-C heatmap of contact matrices for WT and Q140 striatal data at *Htt* locus (top). Colour scale indicates distance between regions in Angstroms (Å). Colour spheres depicting the location of the original baits are shown. In the bottom, insulation score cumulative heatmaps were computed using bins from 4 (40 Kb) to 30 (300 Kb) adding 1 bin each time[37]. Green arrow shows the location of *Htt* at TAD boundary. Black arrow shows the location of subTAD boundary downstream to *Htt*, displaced in Q140 mice data. **d** Left, Manhattan plot representing the distribution of differentially expressed genes (DEGs) in Q140 vs WT striatum at 2 months across the different chromosomes. Significant DEG (adj. *P* value <0.05) are labelled as red dots and the percentage of DEG within each chromosome is shown in peripheral arc. Right, Histogram showing chromosome distribution of DEGs in Q140 vs WT striatum at 2 months of age. Observed numbers were compared with expected numbers for each chromosome and a binomial test (two-sided) was used to assess significant differences, with multiple testing correction using the Bonferroni correction. Source data are provided as a Source Data file.

cells. While it has been reported that shifts in glial cell identity are transcriptional hallmark of human brain aging[43], it is the first time that integrated epigenetic and transcriptional analysis provides evidence for altered maintenance of neuronal identity with age, and additionally shows that this mechanism is early accelerated in a neurodegenerative disease. Our DNA motif analysis indicating enrichment of a sequence recognized by DLX/GATA transcription factors at neuronal-specific regions depleted in H3K27ac in age- and genotype-dependent manners in mouse striatum further support these results, since these transcription factors are critical to the acquisition and maintenance of GABAergic neurons, including striatal neurons[21,22,24]. Strikingly, *Dlx6*, which remains highly expressed in mature GABAergic neurons[44], was decreased both with age and in Q140 vs WT striatum (Supplementary Fig. 11c). This raises the intriguing hypothesis that the HD mutation precipitates age-related neuronal identity loss through a DLX-dependent mechanism.

Specific H3K27ac changes have also been observed in cortical tissues of Alzheimer disease (AD) patients[45], suggesting H3K27ac might be broadly used to investigate epigenetic landscape of neurodegenerative diseases. Since aging is a strong risk factor in AD, it would therefore be important to determine whether H3K27ac reflects epigenetic aging in AD. Supporting this possibility, increasing evidence indicates that enhancers are critical regions implicated in brain diseases, especially since they are hotspots for genetic variations[45–47]. Additional histone modifications may also be age- and/or disease-associated. For example, H4K16ac, a mark enriched at promoters, increases in senescent cells[48]. In humans, normal aging resulted in increased H4K16ac in the lateral temporal lobe, while in AD patients, the trend was opposite, suggesting dysregulation of epigenetic aging[48]. However, it is unclear whether H4K16ac was differentially regulated in neurons and glial cells upon aging. Nevertheless, these results further support the idea of a role for histone acetylation in epigenetic aging and neurodegenerative diseases.

Cell type-specific identity genes are under the control of super-enhancers[3,19,49]. Through chromatin looping, super-enhancers undergo extensive interactions with promoter regions of target genes[27–29]. We hypothesized that decreased H3K27ac at neuronal super-enhancers of HD striatum might associate with disruption of their chromatin architecture, a mechanism that would contribute to downregulation of neuronal identity genes. Our 4C-seq data using HD mouse striatal tissue indicate that, generally, chromatin architecture at super enhancer-regulated genes is not dramatically altered. However, spatial chromatin organization at *Pde10a*, a neuronal identity gene downregulated from early stage in HD striatum, was impaired[50]. The interaction between *Pde10a* promoter and upstream regulatory enhancers was consistently attenuated in the striatum of two HD mouse models, whereas the interaction between *Pde10a* promoter and downstream intronic

region was increased. Our analyses suggest that the HD mutation promotes repressive conformation of chromatin architecture, possibly through a mechanism involving depletion in H3K9me3 and CTCF.

CAG repeat expansion is a dynamic mutation, showing tissue-dependent instability[13]. In HD patients and mouse models, the number of CAG repeats increases over time. Remarkably, somatic CAG repeat expansion is most prominent in the striatum[13,51,52]. Up to 1000 of repeats were measured in post-mortem samples of human patients inheriting <60 repeats[52]. Moreover, recent GWAS studies indicate that gene modifiers of HD onset are enriched in DNA repair genes modulating CAG repeat instability[53,54]. These data strongly support the view that CAG repeat instability contributes to pathogenesis, increasing vulnerability of the striatum[55]. The mechanism may implicate local remodelling of chromatin architecture. In support to this hypothesis, Sun and collaborators recently showed that disease-associated short tandem repeats are located in the vicinity of TAD boundaries. The authors showed that CGG expansion mutation in Fragile X syndrome impairs the insulation between adjacent TADs[14]. Our analyses indicate that CAG expansion in *Htt* gene also lies close to the boundary between two TADs. Using 4C-seq datasets generated across a genomic region encompassing Htt-associated TADs and bioinformatics modelling to build virtual Hi-C maps, we show that CAG expansion mutation in the striatum of HD KI mice affects the insulation of TADs adjacent to disease locus. Moreover, we show that CAG expansion in HD mouse striatum, particularly of young animals, leads to enrichment of genes differentially expressed in chromosome 5, which contains *Htt*. This suggests that CAG expansion locally affects TAD organization and gene regulation. Possible contribution of this mechanism to pathogenesis will remain to be investigated.

In conclusion, we generated high-quality epigenomic datasets to assess the dynamics of epigenetic landscape in neurons and glial cells during early stage of HD progression in mice. Integrating our ChIPseq and 4C-seq datasets with transcriptomic databases, we uncovered that age-related and disease locus-specific mechanisms both contribute to remodelling of chromatin structure in a manner correlating with transcriptional changes. These epigenomic databases, which were generated in a tissue, the striatum, still poorly investigated with omics approaches, represent powerful resource for future studies aimed to decipher brain disease-associated signatures.

## Methods
**Animals**. Heterozygous and homozygous Q140 mice as well as heterozygous R6/1 mice were maintained on C57BL/6J genetic background. All animal studies were conducted in accordance with French regulations (EU Directive 2010/63/UE–French Act Rural Code R 214-87 to 126). The animal facility was approved by veterinary inspectors (authorization no. E6748213) and complies with the Standards for Human Care and Use of Laboratory Animals of the Office of Laboratory

Animal Welfare. All procedures were approved by local ethics committee (CRE-MEAS) and French Research Ministry (no. APAFIS#4301-2016022912385206v2 and no. APAFIS#504-2015042011568820_v3). Mice were housed in a controlled-temperature room maintained on a 12 h light/dark cycle. Food and water were available ad libitum. For molecular analyses (ChIPseq and 4C-seq), mice were killed by cervical dislocation and their striata were rapidly dissected, snap frozen and stored at −80 °C. For 6 months Q140 mice ChIP-seq experiments, tissues from Jackson Laboratory were used. Genotyping was performed by PCR, using tail DNA obtained from 10–15-day-old Q140 mice with primers amplifying the CAG repeat region within the exon 1 of the Huntingtin gene.

**Behavioural tasks.** For actography, spontaneous locomotor activity, reflecting motor function and/or motivation/apathy, was measured during 2 consecutive days; 2-, 6- and 12-month-old animals were tested (2 months, WT: $N = 8$ males, $N = 4$ females, Q140 het: $N = 8$ males, $N = 4$ females; 6 months, WT: $N = 10$ males, Q140 het: $N = 10$ males; 12 months, WT: $N = 6$ males, $N = 4$ females, Q140 het: $N = 6$ males, $N = 4$ females). The number of times that mice disrupt infrared laser were scored and averaged across days and nights. For bar test, motor coordination and balance were assessed using the beam walking assay; 6- and 12-month-old male mice were trained on an elevated narrow beam of 80-cm long, to reach a safe platform containing their home cage. Mice were first habituated to the beam, and then tested through four consecutive trials, each lasting 1 min. The time to cross the beam (latency) was assessed. To assess motor coordination and strength, the accelerating rotarod task was performed; 6- and 12-month-old male mice (6 months, WT: $N = 10$, Q140 het: $N = 10$; 12 months, WT: $N = 10$, Q140: $N = 8$) were trained on a rotarod (Bioseb) at 4 r.p.m. for 2 min. Mice were then tested in three consecutive trials with 45 min inter-trial time, in which the speed of the rod increased from 4 to 40 r.p.m. during 5 min. The latency to fall was recorded as a measurement of mouse performance. This sequence was repeated on 3 consecutive days and values were averaged across trials from the same day. Data were expressed as means ± standard error of the mean (SEM).

**Fluorescence activated nuclear sorting.** Cell-type specific nuclear purification was performed using fluorescent activated nuclear sorting[56,57]. Briefly, frozen striatal tissue was homogenized in ice-cold PBS supplemented with 1× Protease Inhibitors Cocktail (PIC, cOmplete EDTA free, Roche) and cross-linked in 1% formaldehyde for 15 min at room temperature. Cross-linking was stopped by the addition of glycine to final concentration 0.125 M and tissue was washed using ice-cold PBS. Cells were then lysed in Cell Lysis Buffer (10 mM Hepes pH 8; 85 mM KCl; 0.5% NP-40) and nuclei were collected after treatment with Nuclear Extraction Buffer (0.5% SDS, 10 mM EDTA pH 8, 50 mM Tris). Purified nuclei were then resuspended in PBTB (PBS 1×, 5% BSA, 0.5% Tween-20) + 1× PIC, 3% Normal Horse Serum (NHS) and stained using antibody to NeuN (1:1000, Merck Millipore). After washing, nuclei were labelled with Alexa Fluor 488 donkey anti-mouse IgG antibody(1:1500) and washed with ice-cold PBS. Immunostained nuclei were sorted using BD Aria Fusion flow cytometer, recovered in ice-cold 1× PBS, pelleted and stored at −80 °C for posterior ChIP-seq experiments.

**NeuN+/− nuclear quantification using FANS.** Nuclear extracts using the striatum of WT and Q140 heterozygous mice of 2 and 6 months ($n = 4$ per group) were prepared using non-crosslinking nuclear extraction protocol. Briefly, frozen striata were pulverized using a grinder and pestle settle on dry ice and reconstituted in PBS 1× supplemented with 1× PIC. Cell lysis and nuclear extraction were performed by a 10-min incubation in LB1 buffer (1 M HEPES pH 7.5, 5 M NaCl, 0.5 M EDTA pH 8.0, 50% glycerol, 10% NP-40, 10% Triton X-100, 1× PIC) and mechanical dissociation in a glass douncer. Nuclei were then filtered with a 50-µm pore size cell strainer (Sysmex Partec, Kobe, Japan) and stained using Alexa Fluor 405 (AF405) conjugated NeuN antibody (1:200; Novus Biologicals). Nuclear suspension was then sorted using BD FACS ARIA II flow cytometer with a minimum of 30.000 singlet gated events per sample, and NeuN positive (NeuN+) and negative (NeuN−) nuclear proportions were quantified as a relative value of the total number of events registered in these two categories (i.e. "NeuN+" % = "NeuN+"/("NeuN+" + "NeuN−") × 100 and "NeuN−" % = "NeuN−"/("NeuN+" + "NeuN−") × 100).

**NeuN and Sox9 immunohistological analysis.** WT and Q140 heterozygous mice of 2 months ($n = 4$ per group) were intracardially perfused with 4% paraformaldehyde in 0.1 M phosphate buffer, and brains were recovered and post-fixed for additional 6 h at 4 °C. Brains were then cryoprotected by a 48 h incubation in 20% sucrose 0.2 M phosphate buffer, and frozen by a 1-min submersion in dry-ice chilled isopentane. Frozen brains were cut using Leica Microm HM560 cryostat to generate 30-µm-thick striatal coronal sections inter-spaced 150 µm from each other, throughout the full striatum (8–11 slices per mice were generated). Tissue immunostaining was performed as previously described[58]. Briefly, mice coronal sections were washed twice with PBS during 5 min and incubated with NH4Cl 50 mM during 30 min to block free fixation-remaining aldehyde groups and reduce aldehyde-induced tissue auto-fluorescence. The tissue was then permeated during 20 min with a PBS 1× buffer containing 0.5% Triton X-100 and blocked after it for 1 h at room temperature with PBS 1× plus 0.2%, bovine serum albumin, 0.2%

lysine, 0.2% glycine, 0.5% Triton X-100 and 5% normal horse serum. Afterwards, the slices were incubated overnight with neuronal-specific NeuN (1:500, Merck Millipore) and astrocytic-specific Sox9 (refs. [59,60]) (1:200, Cell Signalling) primary antibodies in a buffer containing PBS 1× plus 0.3% Triton X-100, 0.2% bovine serum albumin. Then, brain sections were washed twice in PBS 1x ˣ during 10 min and incubated for 2 h at room temperature in primary antibody buffer with Alexa Fluor 488 donkey anti-mouse IgG or Alexa Fluor 594 donkey anti-rabbit IgG secondary antibodies (1:1500, Invitrogen). Two additional washes of 10 min each with PBS 1× were done to remove the excess of secondary antibodies and nuclear staining was performed by using Hoechst 33258 (1:1000) at r.t. during 5 min. Before mounting the slices, two additional washes with PBS 1× were performed. Afterwards, slices were mounted in glass slides and Mowiol mounting media was used to incorporate glass coverslip. Hamamatsu Nanozoomer Digital Pathology whole slide imaging system (Hamamatsu Photonics) was used for image acquisition of whole brain sections at 40× magnification. The images were then first processed with NDP View v2 software (Hamamatsu Photonics) to delimit the striatal region within each brain section and corresponding area (mm²), and image exportation parameters were established for Sox9 and NeuN channels for the whole experimental image set to avoid any possible bias in posterior counting. The number of NeuN+ and Sox9+ nuclei in delimited striatal area of each brain section was automatically counted, using the spot detector module of ICY software (Institut Pasteur, Paris, France) to avoid experimenter bias[61], and normalized by striatal area (mm²) to compute cell density.

**RNAseq analysis.** RNAseq datasets generated in the striatum of HD KI mice and control mice (GSE65774, https://www.ncbi.nlm.nih.gov/geo/query/acc.cgi?acc=GSE65774)[5] were re-analysed as previously described[3] starting from fastq files. Datasets were downloaded from GEO website according to the GEO identifier provided by the HDinHD website. Tophat2 (ref. [62]) was used for reads mapping using mm10 genome assembly. Quantification of gene expression was performed using HTSeq v0.6.1p1 (ref. [63]), using gene annotations from Ensembl GRCm38 release 87. Read counts were normalized across libraries with the method proposed by Anders and Huber[64]. The method implemented in the DESeq2 (ref. [65]) Bioconductor package (DESeq2_1.14, R_3.3.2) was used to identify significantly DEGs between different mouse genotypes. Resulting $P$-values were adjusted for multiple testing by using the Benjamini and Hochberg method[66]. Down- and up-regulated genes were defined using adj. $P$ val <0.05 and FC < or >1. Top 300-ranked dysregulated genes, based on $P$ val or adj $P$ val, were used in specific analyses. Manhattan plot for genes differentially expressed at 2 months in Q140 vs WT mice and at 6 months in R6/1 vs WT mice were generated using CMplot R package (https://github.com/YinLiLin/R-CMplot), showing −log10(adjpvalue) for all annotated genes. Volcano plots, boxplots and z-score heatmaps were generated using R packages[67]. Cell type-specific striatal RNAseq dataset generated using laser capture microdissected cell populations of WT mouse striatum were analysed as described[10]. Briefly, in this study, the transcriptome of two neuronal populations (i.e. MSNs expressing D1 receptor (D1 MSNs) and MSNs expressing D2 receptor (D2 MSNs), corresponding to neuronal populations affected in HD and predominant in the striatum), and two glial cell populations (astrocytes and microglia), was profiled. To simplify some analyses, D1 and D2 MSNs samples were grouped together and compared to glial samples (i.e. astrocytes and microglia). Down- and upregulated genes in neurons vs glial cells were defined using adj. $P$ val <0.05 and FC < or >1, as described[10].

**Chromatin Immunoprecipitation and sequencing (ChIPseq).** Each ChIP-seq experiment on bulk striatal tissue was performed using the striata of four animals and dividing chromatin extracts in four fractions to allow immunoprecipitating same extract with H3K27ac, H3K27me3 and RNAPII antibodies, and including Input controls. Striata of Q140 heterozygous mice and control wild-type (WT) mice at 2 and 6 months were used in the experiments. ChIPseq data were replicated through four independent experiments (experiment 1, using WT and Q140 striatum at 2 months; experiment 2, using WT and Q140 striatum at 2 months; experiment 3, using WT and Q140 striatum at 6 months; experiment 4, using WT and Q140 striatum at 6 months). Male tissues were used in experiments 1 and 3, and female tissues in experiments 2 and 4. Male and female data of same genotype and age were analysed together to determine differentially enriched regions common to both sexes. Single H3K9me3 ChIPseq experiment was performed using the striatum of Q140 and WT male mice of 6 months. For H3K27ac and H3K27me3 ChIP-seq experiments performed on sorted striatal nuclei of WT mice, 250.000 striatal nuclei were used. The data were replicated through two independent experiments. ChIP-seq was performed as previously described[3] using antibodies to H3K27ac (ab4729,Abcam), H3K27me3 (C15410195, Diagenode), and RNAPII[68]. Briefly, for bulk tissue ChIP-seq experiments, pooled tissues were cut into small fragments, fixed in 1% formaldehyde and incubated for 15 min at room temperature. Cross-linking was stopped by the addition of glycine to final concentration 0.125 M. Tissue fragments were washed with cold PBS supplemented with protease inhibitors. The tissues were then mechanically homogenized in sonication buffer to obtain a homogeneous solution. Tissue homogenates or nuclear suspension (see 'Fluorescence activated cell sorting' in "Methods" section) were sonicated to obtain DNA fragments <500 bp using Covaris Ultrasonicator E220 and centrifuged. The soluble chromatin fraction was pretreated with protein

A Agarose/Salmon Sperm DNA (Millipore) for 45 min at 4 °C. Subsequently, samples were incubated overnight at 4 °C with corresponding primary antibodies. Protein A Agarose/Salmon Sperm DNA was then added and the mixture was incubated for 3 h at 4 °C in a shaker. Agarose beads were washed, protein–DNA complexes were eluted from the beads and de-crosslinked overnight with RNAse A at 65 °C. Proteins were eliminated by 2 h incubation at 45 °C with Proteinase K, and DNA recovered using Qiagen MiniElute PCR Purification Kit.

**ChIPseq library preparation.** ChIP samples were purified using Agencourt AMPure XP beads (Beckman Coulter) and quantified with Qubit (Invitrogen). ChIP-seq libraries were prepared from 2 ng of double-stranded purified DNA using the MicroPlex Library Preparation kit v2 (C05010014, Diagenode), according to manufacturer's instructions. Illumina compatible indexes were added through PCR amplification (7 cycles). Amplified libraries were purified and size-selected using Agencourt® AMPure® XP beads (Beckman Coulter) to remove unincorporated primers and other reagents. Prior to analyses, DNA libraries were checked for quality and quantified using a 2100 Bioanalyzer (Agilent). Libraries were sequenced on Illumina Hiseq 4000 sequencer as Paired-End 50 base reads following Illumina's instructions (IGBMC Genomeast platform). Image analysis and base calling were performed using RTA 2.7.3 and bcl2fastq 2.17.1.14. All ChIP samples successfully went through QC using fastqc (https://www.bioinformatics.babraham.ac.uk/projects/fastqc/).

**ChIPseq analysis: sequence alignment, peak detection and annotation, differential analysis.** Reads were mapped onto Mouse reference assembly GRCm38/mm10 using Bowtie 1.0.0. aligner[69]. Peak detection was performed using SICER[70,71] v1.1 with the following parameters: window size: 200; $e$ value: 0.003. Gap size parameters were selected according to the score value estimated by statistical method implemented in SICER: selected values of gap size are 1000, 600 and 1400 for H3K27ac, RNAPII and H3K27me3, respectively. Peaks were annotated relative to genomic features using Homer AnnotatePeaks v4.9.1 (ref. [72]) with annotation from Ensembl v87. Male and female data were compared using the Bioconductor package ChIPpeakAnno. Inputs were used as controls. ChIPseq data were normalized based on a method that consists in equalizing background regions. Correlative heatmaps and scatter plots were generated for global comparison of samples and replicate analyses. Clustering analysis were performed using seqMINER v1.2.1 (refs. [73,74]) by using Refseq genes of mouse mm10 genome as reference coordinates. Briefly, the samples were normalized to have 1× genome coverage. Reads counts were then compared across samples. For analysis of genotype effect, differential analysis between WT and Q140 samples was performed using SICER. Increased and decreased H3K27ac, RNAPII or H3K27me3 regions were filtered if their adjusted $P$ values $<10^{-5}$, and differential enriched peaks were intersected between replicated experiments and annotated. Independent analyses of male and female samples were also performed using stringent threshold (FDR $<10^{-5}$). Complementary analyses of ChIPseq data were done using the open Galaxy platform GalaxEast (http://www.galaxeast.fr). For analysis of age effect, annotated H3K27Ac, H3K27me3 and RNAPII ChIPseq peaks from independent experiments were intersected to generate a high confidence list of peaks, which were merged across the different biological conditions, for each ChIPseq target. Read coverage was calculated for each sample using bedtools multicov, from BED-Tools[75], and differential enrichment analysis was performed using deseq2[65] with default parameters providing as input normalized reads for each peak and biological sample. To generate neuronal and glial-specific striatal enhancer database, differential analysis between NeuN+ and NeuN− H3K27ac ChIPseq data was performed using SICER. Neuronal- vs glial-specific H3K27ac-enriched regions were selected if their $P$ values $<10^{-15}$. Neuronal- and glial-specific H3K27ac-enriched regions were intersected with H3K27ac regions differentially enriched in Q140 vs WT mice at 2 and 6 months for cell type-specific genotype and age comparisons.

**GO analysis.** GO analysis for multiple datasets comparison was performed using ClusterProfiler package from Bioconductor[76]. List of genes for multiple comparisons were provided as input and enriched Biological Processes terms (FDR <0.05) were identified. For graphical simplification, a semantic similarity simplification was applied using GOSemSim function implemented in clusterProfiler, and top 10 more significant enriched terms were plot for each set of genes in the same plot. Significant biological processes were plotted with a dot size proportional to the gene ratio identified for each term and colour scale according to its adj. $P$ value. Additionally, GO analysis of seqMiner generated clusters was performed using Panther applying filters previously used[77] to reduce redundancy and generalization of selected terms. Independent GO analysis of male and female samples was performed using GREAT[78], selecting top ten terms significantly enriched (FDR <0.05).

**Correlation analysis.** Normalized read values of annotated peaks assigned to neuronal- or glial-specific enhancers were used to compute Spearman's correlation coefficient, using ggpubr R package. For cell-type specific analysis of regions changing in age- and genotype-dependent manners, peaks differentially enriched in H3K27ac ($P$ <0.05) between Q140 and WT mice at 2 months and between 6 and 2 months in WT mice were filtered and defined as Non-specific, Neuronal-specific

or Glial-specific, through intersection with neuronal- and glial-specific enhancer database. Marginal distribution density and scatter plots were simultaneously generated, combining geom density tool from cowplot and ggscatter from ggpubr R packages plotting and using normalized read values obtained from DESeq2. Z-score hierarchical heatmap generation was performed using ClustVis[79].

**DNA motif analysis.** Motif analysis at neuronal- and glial-specific enhancers showing age- and genotype-dependent changes in H3K27ac was performed using RSAT[79,80]. Differentially enriched peaks ($P$ <0.05) were categorized as Non-specific, Neuronal-specific or Glial-specific and motif analysis using standard parameters was performed in Neuronal vs Non-specific and Glial vs Non-specific regions. Top discovered motifs ($e$-value <$10^{-4}$) associated to transcription factors found in Hocomoco (version 11) human and mouse PWMs database were selected.

**Network analysis.** Network analysis of genes linked to neuronal- and glial-specific enhancers showing changes in H3K27ac in age- and genotype-dependent manners was performed using STRING[81,82]. Annotated genes from differentially enriched peaks ($P$ <0.05) were selected and categorized in Neuronal- or Glial-specific and protein–protein interaction (PPI) network analysis was conducted using default parameters from STRING. PPI score was used as a measure of co-regulatory network complexity analysis.

**Circular chromatin conformation capture (4C-seq).** 4C-seq was performed as previously described[83] with slight modifications. 4C-seq data were replicated for each condition, using male tissues in a first replicate and female tissues in the second replicate. Briefly, 5 million nuclei per biological condition were purified using same steps described in the FANS approach[56,57], except that percentage of formaldehyde was increased up to 2%. Purified nuclei were digested O/N with the first restriction enzyme (DpnII, New England Biolabs) and posteriorly subjected to an overnight ligation with T4 DNA Ligase (New England Biolabs). Subsequently, chromatin was de-crosslinked and purified after proteinase K and RNAse A treatment. A second digestion using Csp6I (Thermo Scientific) was performed O/N, followed by final O/N ligation with T4 DNA Ligase and DNA purification using phenol/chloroform extraction after sample volume reduction by using 90% under-saturated phenol (UPT phenol). The resultant 4C DNA template was used to generate 4C-seq libraries by performing a PCR (Long Template PCR system, Roche) with target-specific designed reading and non-reading primers (see table below) containing Illumina sequencer adapters. For primer design, a region surrounding the TSS of the gene of interest (±2 kb) was retrieved and primers were designed using SnapGene (v.1.1.3) for regions that fulfilled the following criteria: distance between DpnII restriction site and the consecutive Csp6I restriction site >350 bp; distance between DpnII restriction site and the following DpnII restriction site after Csp6I >500 bp and <1500 bp (Supplementary Table 1). A primer validation step was included to verify their specificity. Then, generated 4C-seq libraries were purified with SPRI select beads (Beckman) to discard primer dimer DNA products and 4C-seq DNA template were quantified using Bioanalyzer and pooled equimolarly for sequencing using 50 bp single-end Hiseq 4000 sequencer (IGBMC Genomeast platform).

**4C-seq data analysis.** Reads were mapped to mm10 with Bowtie[69], then processed and visualized with 4See (refs. [53,84]). At the *Pde10a* locus, interactions were called on individual male and female samples with peakC[85] using default parameters, including a window size of 21 fragments. To assess differential interactions at the reproducibly called interacting regions, two-tailed $t$ tests were performed on the mean values of the 4See-outputted quantile-normalized scores of the fragments contained within each interacting region. CTCF data from mESC was retrieved from GSE125129 GEO datasets (https://www.ncbi.nlm.nih.gov/geo/query/acc.cgi?acc=GSE125129).

**Virtual Hi-C (vHi-C) data generation and analysis.** The 3D chromatin models representing the *Htt* region using the different 4C-seq datasets generated on WT and Q140 Homozygous striatal samples were built using 4Cin[38]. The locus modelled is comprised within chromosome 5: 33954729-36029316 of mm10 mouse genome. Default parameters of the program were used except for the number of fragments that each bead represents, which was set to 25. Chromatin painting was performed using H3K27ac, H3K27me3 and RNAPII ChIP-seq data of corresponding biological conditions using paint_model.py script from 4C-in. To identify TAD boundaries using the models, an insulation score was computed as previously described[39].

**Statistics.** Mice with the same age and sex were randomly allocated to the different experimental groups. Blinding was applied to behavioural experiments. For bar plots, centred regions indicate the mean ± sem; for boxplots, centred regions indicate the median, box limits, upper and lower quartiles and whiskers, 1.5× interquartile range. All measurements were taken from distinct samples. No data were excluded from analyses. For pairwise comparisons of average, data were tested for normality using the Shapiro's test. Statistical analyses included two-tailed, paired or unpaired, Student's $t$ test, one-way analysis of variance. In case the

samples were significantly non-normal, non-parametric tests, including Kruskal–Wallis and binomial tests were performed. For multiple comparisons, the Newman-Keuls test or Benjamini-Hochberg method was applied. $P$ values < 0.05 were considered to be statistically significant, except when otherwise indicated. No statistical method was used to predetermine sample size, but our sample sizes are based on similar, previously established, experimental designs.

**Reporting summary**. Further information on experimental design is available in the Nature Research Reporting Summary linked to this paper.

## Data availability

The datasets generated in this study are available at NCBI GEO under the following accession numbers: "GSE144684" and "GSE144699". A reporting summary for this Article is available as a Supplementary Information file. All other relevant data are available within the Article, Supplementary Information, or available from the author upon request. Source data are provided with this paper.

## Code availability

Custom codes are available from the corresponding author upon reasonable request.

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

## Acknowledgements

We thank PsychoGenics for providing tissues as part of a contract research agreement with CHDI. We thank O. Bildstein, D. Egesi, G. Edomwonyi and C. Strittmatter (LNCA UMR7364) for assistance in animal care. Sequencing was performed by the GenomEast Platform, a member of the 'France Génomique' consortium (ANR-10-INBS-2009). We thank C3UPO for the high-performance computing (HPC) support at Uni. Pablo de Olavide. This study was supported by CHDI foundation, Inc, the Agence Nationale de la Recherche (ANR-2017-CE12-0027), the Centre National de la Recherche Scientifique (CNRS) and the University of Strasbourg. R.A.V. was supported by post-doctoral fellowship from CHDI. J.S. and A.B. were bioinformatician and technician supported by CHDI. C.L. and A.A. were recipients of doctoral fellowships from the French government and the Association Huntington France (AHF), respectively. Work in the T.S. group is supported by the European Research Council (ERC) under the European Union's Horizon 2020 research and innovation program (Starting Grant 678624 - CHROM-TOPOLOGY), the ATIP-Avenir program, and the grant ANR-10-LABX-0030-INRT, a French State fund managed by the Agence Nationale de la Recherche under the frame program Investissements d'Avenir ANR-10-IDEX-0002-02. A.M.M. was supported by funds from INCA. J.-L.G.-S. was supported by the Spanish government (grant no. BFU2016- 74961-P) and the institutional grant Unidad de Excelencia María de Maeztu (no. MDM-2016-0687). I.I.-A. was supported with a FEBS long-term fellowship.

## Author contributions

R.A.V. designed and performed the experiments, analysed the data, and wrote and edited the manuscript. J.S. analysed the data, and wrote and edited the manuscript, C.L. performed the experiments, analysed the data and edited the manuscript, A.M.M. assisted with experiments and data analysis, and edited the manuscript, I.I.A. analysed the data and edited the manuscript, A.A. performed the experiments and edited the manuscript, N.K. assisted with data analysis and edited the manuscript, A.B. performed the experiments and edited the manuscript, B.C. performed the experiments and edited the manuscript, J.L.G.S. interpreted the experiments and edited the manuscript, J.C.C. interpreted the experiments and edited the manuscript, A.L.B. interpreted the experiments and edited the manuscript, T.S. analysed the data, and wrote and edited the manuscript. K.M. designed the experiments, analysed the data, and wrote and edited the manuscript.

## Competing interests

The authors declare no competing interests.
