## [Peer Review File · Nature Communications]

REVIEWER COMMENTS

Reviewer #1 (Remarks to the Author):

Others previously showed that striatal epigenetic and transcriptional changes are present in HD mouse models and HD patient tissue. Many of these studies utilized relatively late tissue for study, particularly those using patient material. In this study the investigators use chromatin IP and RNAseq analyses to examine early genomic changes in the Q140 knockin HD mouse, taking advantage that this is a slow progressing model of HD so that genomic signatures at an early disease stage can be more readily assessed. Importantly, state-of-the-art molecular and computational seemed to have been used. While previous work has linked epigenetic alterations in HD with those found in aging brain, this study nicely demonstrates that the enhanced age-related epigenetic changes occur very early in a striatum expressing mutant HTT – well before neuronal pathology and motor deficits. Moreover, by integrating chip-seq data with cell-specific transcriptomic data from the striatum they show that in mutant mice, age-dependent effects in at both neuronal and glial-specific enhancers. Lastly, data on the effect of an expanded CAG repeat on local chromatin 3D structure are reported. My concern with this manuscript is one common to most studies reporting extensive amount of genomic data, the figure are very dense. For example in Figure 1 with all of the data included, the critical finding on neuronal vs glial gene ontology shown in Fig 1C is visually minimized. I suggest that authors spend further effort on moving more of the results/images to supplementary figures so that the figures in the main text are focused more on their key findings.

Reviewer #2 (Remarks to the Author):

The manuscript by Alcalá-Vida and colleagues investigates the relationship between changes in transcription, histone modifications and chromatin interactions associated with Huntington's disease (HD) in striatal neurons. The multi-omic analysis is conducted in the slow progressing HD knock-in mouse model and focused on the early stages of pathology (2 and 6 months). The genomic data generated here are of high quality and should represent a useful resource for future studies on the pathoetiology of HD. In addition to this general value, the study provides some relevant and novel findings, including the demonstration of acceleration of age-related transcriptome and epigenome changes in HD, and the identification of specific loci in which the 3D chromatin architecture is altered. The most relevant of such loci is Htt itself, pinpointing a direct role for the CAG expansion in the pathology (which interestingly would be independent of the production of aberrant transcripts and proteins). Overall, this is an interesting piece of work that extends our current understanding of the molecular basis of HD.

I have some comments that could help the authors to improve their manuscript:

1. My main concern refers to the interpretation of the data in terms of changes in "neuronal and glial identities". Genotype-based screens were performed using bulk striatal chromatin; the authors very effectively use the information from NeuN+ vs NeuN- comparisons performed in wild type mice to dissect the complex signal from bulk tissue, but still the original signal proceeds of a mix of neuronal and non neuronal cells. The approach used by the authors would be valid if there is no difference in the cellular composition of the tissue, but this may not be the case in their model. The authors should present evidence that there is no significant neuronal loss or gliosis at the two time points investigated. Given the importance for the interpretation of the results, the sentence in the introduction indicating that the HD model "shows limited neuronal death, even at late disease stage (ref15)" seems insufficient. I would be particularly interested in the quantification of gliosis. It is possible (and it fits quite well with their results) that the downregulation and H3K27 hypoacetylation of neuronal genes really reflects an attenuation of "neuronal identity". However, the upregulation and H3K27 hyperacetylation of glial genes might

reflect instead an increase in the presence of glial cells. Immunohistological analyses can easily address this concern. In addition, the authors could also quantify the relative abundance of NeuN+ and NeuN- in Q140 mice when compared to controls. The authors may need to revise several sentences of the text depending on the result of these experiments (although the main conclusions are unlikely to be affected).

2. The description of the dataset is misleading. The authors refer to "replicates", but they did not perform real biological replicates. They obtained a sample from males and another from females. The sample size per condition (genotype and HPTM) is only 1. Apparently, they did not consider "sex" as a variable in their analyses and focused exclusively on genotype-related changes. The authors should clearly explain this decision and how sex was considered in their analyses. They could also indicate what percentage of HD-related changes presents a sex-genotype interaction. A very obvious example of the impact of sex on their analyses can be found in the scatter plots presented in Supp. Figure S2c. The dot clouds observed outside the diagonal in the H3K27me3 graphs must correspond to genes in the X chromosome (since one of the two alleles would be decorated with this mark in female chromatin). "Rep1" seems to correspond to females in the 2 months samples, while "Rep1" should correspond to the males in the 6 months samples.

3. The authors have apparently generated profiles for H3K9me3 (Fig. 4). Why are these profiles presented so late in the article? It would have made sense to present them earlier, in Figure 1, together with the H3K27 modifications. I understand that they only obtained samples for 6-month old males and they do not have any "replicate" in this case.

4. The access number GSE14469 does not correspond to 4Cseq data, but to a completely unrelated study on synovial sarcoma-like tumors.

5. Was the 4C dataset in R6/1 mice generated in the context of this study? Since the authors present RNA-seq data of R6/1 mice in Supp. Figure S9d, they could also prepare a circus graph for this strain similar to the one presented in Fig. 5d for Q140 mice. It would be interesting to confirm the disappearance of the enrichment in chromosome 5. Moreover, maybe they will detect enrichment in a different chromosome coinciding with the insertion site for the mHtt transgene.

6. Do the authors have any hypothesis concerning the specificity of 3D changes? Why is Pde10a affected but not other loci?

7. Genomic snapshot in Figure 1 and others are missing the scales both in the X (length/distance) and Y (signal/reads density) axes. Also, the information about gene position and structure (exon/introns, transcript variants) is impossible to read. These panels should be adapted to a printable format.

8. In general, the authors should revise all the figures (main and supplementary figures) to eliminate unreadable text. Font size in some figures is often too small. They should also try to homogenize the text presented in the figures, both in style and size. In particular, the values on the Y-axis in numerous graphs should be enlarged.

Minor comments:

9. The authors should revise the abstract. The text seems too technical and descriptive and does not clearly highlight the main biological findings of the study. In that sense, the last paragraph of the introduction does, in my opinion, a better work.

10. They should also revise the wording of some sentences to clarify the message or accuracy. For example:

- P.6: "... , normal age-dependent transcriptional regulation...was enhanced..."
- P.6: "... particularly prone to depletion...with age"
- P.7: "...age-related reprogramming of neuronal and glial identity genes"

- P.8: "... were specific to the striatum" (specificity is not shown, only the difference with stem cells).
- P.10: "...age-related acceleration of ... reprogramming of neuronal and glial identities"...

REVIEWER COMMENTS

Reviewer #1 (Remarks to the Author):

Others previously showed that striatal epigenetic and transcriptional changes are present in HD mouse models and HD patient tissue. Many of these studies utilized relatively late tissue for study, particularly those using patient material. In this study the investigators use chromatin IP and RNAseq analyses to examine early genomic changes in the Q140 knockin HD mouse, taking advantage that this is a slow progressing model of HD so that genomic signatures at an early disease stage can be more readily assessed. Importantly, state-of-the-art molecular and computational seemed to have been used. While previous work has linked epigenetic alterations in HD with those found in aging brain, this study nicely demonstrates that the enhanced age-related epigenetic changes occur very early in a striatum expressing mutant HTT – well before neuronal pathology and motor deficits. Moreover, by integrating chip-seq data with cell-specific transcriptomic data from the striatum they show that in mutant mice, age-dependent effects in at both neuronal and glial-specific enhancers. Lastly, data on the effect of an expanded CAG repeat on local chromatin 3D structure are reported. My concern with this manuscript is one common to most studies reporting extensive amount of genomic data, the figure are very dense. For example in Figure 1 with all of the data included, the critical finding on neuronal vs glial gene ontology shown in Fig 1C is visually minimized. I suggest that authors spend further effort on moving more of the results/images to supplementary figures so that the figures in the main text are focused more on their key findings.

We thank reviewer 1 for his constructive comments and suggestions. To improve the presentation of the data, we have moved previous Fig. 1a to supplementary figure, and have increased the size of Fig. 1c (now Fig. 1b). Also, we have moved several main results into supplementary figures. Overall, the reorganization has led to three additional supplementary figures. Specifically:

Previous Fig.1a was moved to Supplementary Fig. 2

Previous Fig.2a (middle and right graphs) was moved to Supplementary Fig.6a

Previous Fig.2d was moved to Supplementary Fig.7b

Previous Fig.3e,h,i were moved to Supplementary Fig. S11a,b,c

Reviewer #2 (Remarks to the Author):

The manuscript by Alcalá-Vida and colleagues investigates the relationship between changes in transcription, histone modifications and chromatin interactions associated with Huntington's disease (HD) in striatal neurons. The multi-omic analysis is conducted in the slow progressing HD knock-in mouse model and focused on the early stages of pathology (2 and 6 months). The genomic data generated here are of high quality and should represent a useful resource for future studies on the pathoetiology of HD. In addition to this general value, the study provides some relevant and novel findings, including the demonstration of acceleration of age-related transcriptome and epigenome changes in HD, and the identification of specific loci in which the 3D chromatin architecture is altered. The most relevant of such loci is Htt itself, pinpointing a direct role for the CAG expansion in the pathology (which interestingly would be

independent of the production of aberrant transcripts and proteins). Overall, this is an interesting piece of work that extends our current understanding of the molecular basis of HD.

I have some comments that could help the authors to improve their manuscript:

We thank reviewer 2 for his constructive comments and suggestions.

1. My main concern refers to the interpretation of the data in terms of changes in “neuronal and glial identities”. Genotype-based screens were performed using bulk striatal chromatin; the authors very effectively use the information from NeuN+ vs NeuN- comparisons performed in wild type mice to dissect the complex signal from bulk tissue, but still the original signal proceeds of a mix of neuronal and non neuronal cells. The approach used by the authors would be valid if there is no difference in the cellular composition of the tissue, but this may not be the case in their model. The authors should present evidence that there is no significant neuronal loss or gliosis at the two time points investigated. Given the importance for the interpretation of the results, the sentence in the introduction indicating that the HD model “shows limited neuronal death, even at late disease stage (ref15)” seems insufficient. I would be particularly interested in the quantification of gliosis. It is possible (and it fits quite well with their results) that the downregulation and H3K27 hypoacetylation of neuronal genes really reflects an attenuation of “neuronal identity”. However, the upregulation and H3K27 hyperacetylation of glial genes might reflect instead an increase in the presence of glial cells. Immunohistological analyses can easily address this concern. In addition, the authors could also quantify the relative abundance of NeuN+ and NeuN- in Q140 mice when compared to controls. The authors may need to revise several sentences of the text depending on the result of these experiments (although the main conclusions are unlikely to be affected).

We agree with the concern of the reviewer regarding the possibility of a change in the cellular composition of neuronal and glial cell populations in the striatum of HD Q140 mice, which could underlie the cell-type specific epigenomic alterations we observe. As nicely suggested by the reviewer, we have addressed this concern using two different strategies, Fluorescence-Activated Nuclear Sorting (FANS) and immunohistological analysis. Using FANS, we have quantified the relative abundance of NeuN+ and NeuN- in the striatum of Q140 and WT mice at 2 and 6 months of age. The results show that NeuN+ / NeuN- ratio remains stable across genotypes and ages, indicating that there is no significant age-dependent neuronal loss and/or gliosis in the HD context (at least until 6 months). In addition, we have performed immunohistological analyses on striatal slices of Q140 and WT mice at 2 months of age, when cell-type specific signature was particularly prominent, using NeuN and Sox9 antibodies to quantify neurons and astroglial cells (new Supplementary Fig. 5). Our results also indicate that the density of neurons and astroglial cells is comparable between Q140 and WT animals. Thus, these data support the view that early depletion in H3K27ac at neuronal genes and increased H3K27ac at glial genes in Q140 striatum do not result from neuronal loss and/or gliosis, but rather reflect cell-autonomous epigenetic changes, which strengthens our conclusion. These results are described in the manuscript: “Remarkably, the effect was more specific at 2 vs 6 months (Fig. 1d,e and Supplementary Fig. 4e) and did not result from neuronal loss and/or astroglial cells, since the relative abundance of neuronal vs non-

neuronal populations (including astrocytes) were comparable between the striatum of Q140 and WT mice (Supplementary Fig. 5)".

2. The description of the dataset is misleading. The authors refer to "replicates", but they did not perform real biological replicates. They obtained a sample from males and another from females. The sample size per condition (genotype and HPTM) is only 1. Apparently, they did not consider "sex" as a variable in their analyses and focused exclusively on genotype-related changes. The authors should clearly explain this decision and how sex was considered in their analyses. They could also indicate what percentage of HD-related changes presents a sex-genotype interaction. A very obvious example of the impact of sex on their analyses can be found in the scatter plots presented in Supp. Figure S2c. The dot clouds observed outside the diagonal in the H3K27me3 graphs must correspond to genes in the X chromosome (since one of the two alleles would be decorated with this mark in female chromatin). "Rep1" seems to correspond to females in the 2 months samples, while "Rep1" should correspond to the males in the 6 months samples.

Male and female animals were indeed used to generate independent ChIPseq datasets. It is noteworthy that male and female datasets were not generated at the same time. Since n=1 per sex, genotype and HPTM, we could not distinguish between sex and batch effects. Therefore, we did not consider sex as a variable. We believe that considering male and female datasets as biological replicates is an acceptable approximation because HD has not been described as sex-dependent. This precision is now specified in the materials and section methods: "Male tissues were used in experiments 1 and 3, and female tissues in experiments 2 and 4. Male and female data generated at same age were considered as biological replicates. Since HD is not a sex-dependent disease, we consider that the approximation is reasonable."

We thank the reviewer for pointing to the mistake we did inverting some graphs using rep1 and rep2 datasets: now rep1 always corresponds to male datasets and rep2 to female datasets. Graphs in supplementary Supplementary Fig. 3 have been modified accordingly.

3. The authors have apparently generated profiles for H3K9me3 (Fig. 4). Why are these profiles presented so late in the article? It would have made sense to present them earlier, in Figure 1, together with the H3K27 modifications. I understand that they only obtained samples for 6-month old males and they do not have any "replicate" in this case.

As commented by the reviewer, H3K9me3 ChIP-seq data were exclusively generated at 6 months of age (and using male samples only). Since the data are not used for integrated analysis shown in Supplementary Fig.2, they are not introduced at that stage. The generation of this dataset is now better specified in the material and method section "Single H3K9me3 ChIPseq experiment was performed using the striatum of Q140 and WT male mice of 6 months". Moreover, we have included profiles reflecting the quality of the datasets in new Supplementary Fig. 12.

4. The access number GSE14469 does not correspond to 4Cseq data, but to a completely unrelated study on synovial sarcoma-like tumors.

There was indeed a mistake in the GEO number for 4Cseq data, the right access number is GSE144699. This has been corrected.

5. Was the 4C dataset in R6/1 mice generated in the context of this study? Since the authors present RNA-seq data of R6/1 mice in Supp. Figure S9d, they could also prepare a circus graph for this strain similar to the one presented in Fig. 5d for Q140 mice. It would be interesting to confirm the disappearance of the enrichment in chromosome 5. Moreover, maybe they will detect enrichment in a different chromosome coinciding with the insertion site for the mHtt transgene.

R6/1 4C-seq data were generated in the context of the study to compare with Q140 4Cseq data. As suggested by the reviewer, we have generated a circus graph showing the distribution of differentially expressed genes (DEG) in R6/1 vs WT striatum across the different chromosomes, using RNAseq data we generated in previous study (Achour et al. 2015). This new graph (Supplementary Fig. 13d) shows that chromosome 5 is not enriched in DEG in R6/1 mice. We have described it in the results: « As expected, chromosome 5 was not enriched in DEG in the striatum of R6/1 transgenic mice overexpressing CAG-expanded *HTT* exon-1 vs WT mice ^{28,38}, and *Htt* and *Grk4* were unchanged in the striatum of R6/1 mice (Supplementary Fig. 13d,e) ».

6. Do the authors have any hypothesis concerning the specificity of 3D changes? Why is Pde10a affected but not other loci?

The question addressed by the reviewer is of great interest, but difficult to address. The direct link between histone modifications and chromatin topology appears to be anything but trivial, with topological context-dependent effects and still poorly understood mechanisms governing chromatin loops formation and maintenance. Our analyses suggest that chromatin architecture and transcription at Pde10a could be regulated by a repressive loop involving a CTCF site in Pde10a intronic region, which may be stabilized in the HD context. Such a mechanism may only affect subsets of neuronal identity genes, and could result from local loss of H3K9me3, promoting on-site recruitment of CTCF, which might explain why chromatin architecture at Pde10a, and not at the other neuronal identity genes tested (i.e. *Gpr6* and *Ptpn5*), was altered in the striatum of Q140 mice. This is now specified in the results section: “Altogether, these results suggest that chromatin topology is largely unchanged by H3K27ac depletion at neuronal super enhancers or transcriptional down-regulation of their target genes during HD onset, although locus-specific architectural changes involving additional mechanisms may be observed at subsets of neuronal super enhancers. »

7. Genomic snapshot in Figure 1 and others are missing the scales both in the X (length/distance) and Y (signal/reads density) axes. Also, the information about gene position and structure (exon/introns, transcript variants) is impossible to read. These panels should be adapted to a printable format.

Scales for snapshots have been included when missing, information about gene position and structure have been changed to improve readability. We have changed concerned figures accordingly (i.e. new Fig. 1a,b, 2a,b, 4, 4, supplementary Fig.12)

8. In general, the authors should revise all the figures (main and supplementary figures) to eliminate unreadable text. Font size in some figures is often too small. They should also try to homogenize the text presented in the figures, both in style and size. In particular, the values on the Y-axis in numerous graphs should be enlarged.

The figures have all been checked and modified to improve their readability, when necessary.

Minor comments:

9. The authors should revise the abstract. The text seems too technical and descriptive and does not clearly highlight the main biological findings of the study. In that sense, the last paragraph of the introduction does, in my opinion, a better work.

The abstract has been changed to better highlight biological findings:

« Temporal dynamics and mechanisms underlying epigenetic changes in Huntington's disease (HD), a neurodegenerative disease primarily affecting the striatum, remain unclear. Using slow progressing knockin mouse model, we have profiled HD striatal epigenome at two early disease stages. Data integration with cell type-specific striatal enhancer and transcriptomic databases demonstrates acceleration of age-related epigenetic remodeling and transcriptional changes at neuronal- and glial-specific genes from prodromal stage, before the onset of motor deficits. Also, 3D chromatin architecture, while generally preserved at neuronal enhancers, was altered at disease locus. Specifically, the HD mutation, a CAG expansion in the *Htt* gene, locally impaired spatial organization of the chromatin and the regulation of proximal genes. Thus, our data provide evidence for two early and distinct mechanisms underlying chromatin structure changes in HD striatum, correlating with transcriptional changes: the HD mutation globally accelerates age-dependent epigenetic and transcriptional reprogramming of brain cell identities, and locally affects 3D chromatin organization. »

10. They should also revise the wording of some sentences to clarify the message or accuracy. For example:

- P.6: "..., normal age-dependent transcriptional regulation....was enhanced..."

The sentence was changed for:

"Importantly, physiological age-dependent transcriptional changes of neuronal- and glial-specific genes were accelerated by the HD mutation.."

- P.6: "... particularly prone to depletion...with age"

The sentence was changed for:

"These results suggest that neuronal-specific genes are particularly prone to reduced H3K27ac occupancy and RNAPII recruitment with age"

- P.7: "...age-related reprogramming of neuronal and glial identity genes"

The sentence was changed for:

"...thereby resulting in acceleration of age-related regulation of neuronal and glial identity genes."

- P.8: "... were specific to the striatum" (specificity is not shown, only the difference with stem cells).

The sentence was changed for:

« Furthermore, upstream interacting regions were not observed in embryonic stem cells (mESC)²⁹, suggesting a role in cell type-specific expression of *Pde10a*»

• P.10: "...age-related acceleration of ... reprogramming of neuronal and glial identities"...

The sentence was changed for:

« Thus, we provide for the first time evidence that the HD mutation leads to acceleration of age-related transcriptional and epigenetic regulation of neuronal and glial cell identities in the striatum. »

REVIEWER COMMENTS

Reviewer #2 (Remarks to the Author):

The revised manuscript is significantly improved and addressed most of my criticisms. In general, the figures are much more accessible and easier to read. However, there is a point that I think still requires additional attention.

As indicated in point 2 in my review, I found the description of the datasets misleading because the authors referred to samples obtained from male or female mice as "replicates". The authors confirmed this point and responded that since the male and female datasets were not generated at the same time, they cannot distinguish between sex and batch effects, which is true, but I do not think that justifies ignoring the sex/batch effect. They should take this in consideration in their statistical analysis and likely increase the stringency of the screen considering these extra variables. Contrary to the authors, I do not believe "that considering male and female datasets as biological replicates is an acceptable approximation". I would prefer that the authors clearly explain their strategy in the main text (for example, the first reference to sex differences in line 283 cannot be understood without an explanation that can be only found in the Methods section) and change the label "rep1" to "male" and the label "rep2" to "female". They should conduct independent analyses in the samples for each sex and highlight the findings observed in both cases. They should also indicate what HD-related changes may present a sex-genotype interaction. Although HD is not sex-dependent, there are several studies describing differences in the progression of the disease (see very recent review by Zielonka and Stawinska-Witoszynska), including sex differences in the 140 CAG knock-in mice used in this study (Dorner et al., 2007). Are all the main conclusions, regarding acceleration of age-dependent epigenetic and transcriptional changes and 3D chromatin organization, supported by evidence in both sexes?

REVIEWER COMMENTS

Reviewer #2 (Remarks to the Author):

The revised manuscript is significantly improved and addressed most of my criticisms. In general, the figures are much more accessible and easier to read. However, there is a point that I think still requires additional attention.

As indicated in point 2 in my review, I found the description of the datasets misleading because the authors referred to samples obtained from male or female mice as “replicates”. The authors confirmed this point and responded that since the male and female datasets were not generated at the same time, they cannot distinguish between sex and batch effects, which is true, but I do not think that justifies ignoring the sex/batch effect. They should take this in consideration in their statistical analysis and likely increase the stringency of the screen considering these extra variables. Contrary to the authors, I do not believe “that considering male and female datasets as biological replicates is an acceptable approximation”. I would prefer that the authors clearly explain their strategy in the main text (for example, the first reference to sex differences in line 283 cannot be understood without an explanation that can be only found in the Methods section) and change the label “rep1” to “male” and the label “rep2” to “female”. They should conduct independent analyses in the samples for each sex and highlight the findings observed in both cases. They should also indicate what HD-related changes may present a sex-genotype interaction. Although HD is not sex-dependent, there are several studies describing differences in the progression of the disease (see very recent review by Zielonka and Stawinska-Witoszynska), including sex differences in the 140 CAG knock-in mice used in this study (Dorner et al., 2007). Are all the main conclusions, regarding acceleration of age-dependent epigenetic and transcriptional changes and 3D chromatin organization, supported by evidence in both sexes?

We thank the reviewer for his supportive comments, and address the remaining issue. We agree with the reviewer that even if HD is not described as a sex-dependent disease, recent papers support sex-dependent effects (e.g. Zielonka and Stawinska-Witoszynska 2020 and Dorner and collaborators, 2007). We used samples from male and female animals, rather than samples from only one sex, to avoid sex-dependent bias in our analyses, and we designed our experiments to enable the capture of HD-related epigenetic changes common to both males and females. Our analyses were conducted to find common events in male and female samples. Thus, our main conclusions regarding acceleration of age-dependent epigenetic and transcriptional changes and 3D chromatin organization are based on these analyses: they are supported by evidence in both sexes. We were aware that our data might not catch sex-dependent changes, due to batch effects and/or low number of samples per sex.

1) To make it clearer, we now explain our strategy at the beginning of the Results section “HD is generally described as a sex-independent disease affecting similarly males and females, though recent studies indicate sex-dependent effects influencing disease progression (Zielonka and Stawinska-Witoszynska 2020, Dorner et al. 2007). To avoid sex-dependent bias in our analyses, we used striatal tissue from both male and female mice. ChIPseq experiments using Q140 and WT samples of specific age and sex were performed simultaneously. For practical reasons, experiments performed on different

sexes and at different ages were conducted at different times (see Methods). The data were of high quality, as shown by peak enrichment signal to noise rates, correlation analyses and additional quality analyses (Fig. 1a, Supplementary Fig. 2,3). Moreover, principal component analysis (PCA) showed that sample variability was essentially explained by age (H3K27ac, RNAPII, H3K27me3), batch/sex (H3K27ac, RNAPII, H3K27me3) and genotype (H3K27ac) (Supplementary Fig. 3). Since sex and batch effects could not be distinguished, we focused our analyses on genotype- and age-dependent changes, analyzing together male and female samples to assess epigenetic changes common to both sexes.” Thus, PCA analysis was moved to Supplementary Fig.3.

2) We have removed the sentence in the Methods “...that considering male and female datasets as biological replicates is an acceptable approximation” and wrote instead: “Male and female data of same genotype and age were analysed together to determine differentially enriched regions common to both sexes.”

3) We have removed ‘Rep 1’ and ‘Rep2’ in concerned figures and use instead ‘male’ and ‘female’

4) We have performed independent analyses of male and female samples using stringent threshold ($FDR < 10^{-5}$), and have performed gene ontology analysis on those analyses. The results are presented in new supplementary Fig. 5 & 6, and described in the Results section: “Independent analysis of male and female ChIPseq samples supported the results of combined analysis of male and female samples (Supplementary Fig. 5,6). Notably, H3K27ac was early depleted and enriched at neuronal and glial genes, respectively (Supplementary Fig. 5).”

5) We make clearer in the results section the fact that we used male and female striatal samples in our 3D chromatin architecture experiments, first in the paragraph addressing 3D chromatin architecture at striatal identity genes: “Experiments were performed using the striatum of male and female Q140 and WT mice at 6 months, and targeting super enhancer-regulated genes down-regulated in HD striatum, including *Pde10a*, *Gpr6* and *Ptpn5*, and non-super enhancer-regulated genes such as *Msh2*, as a control (Fig. 4a,b and Supplementary Fig. 15,16a-c)”. Chromatin looping at *Pde10a* was impaired in male and female Q140 samples (Fig. 4b and Supplementary Fig. 15).” We include an additional figure (supplementary figure 15, showing male and female sample display similar 4Cseq profiles at *Pde10a* locus. Second we specify that we use male and female samples in the paragraph addressing the effect of CAG expansion on 3D chromatin architecture: « Striatal tissues of male and female animals were used in the analysis. »

REVIEWERS' COMMENTS

Reviewer #2 (Remarks to the Author):

The authors have effectively addressed my last concerns.